# Systematic multiscale models to predict the compressive strength of fly ash-based geopolymer concrete at various mixture proportions and curing regimes

**Hemn Unis Ahmed[1], Ahmed Salih Mohammed**![ORCID][1]*****, **Azad A. Mohammed[1], Rabar H. Faraj[2]**

**1** Department of Civil Engineering, College of Engineering, University of Sulaimani, Sulaimaniyah, Iraq,
**2** Civil Engineering Department, University of Halabja, Halabja, Kurdistan Region, Iraq

* ahmed.mohammed@univsul.edu.iq

**Data Availability Statement:** All relevant data are within the manuscript.

**Funding:** The authors received no specific funding for this work.

## Abstract

Geopolymer concrete is an inorganic concrete that uses industrial or agro by-product ashes as the main binder instead of ordinary Portland cement; this leads to the geopolymer concrete being an eco-efficient and environmentally friendly construction material. A variety of ashes used as the binder in geopolymer concrete such as fly ash, ground granulated blast furnace slag, rice husk ash, metakaolin ash, and Palm oil fuel ash, fly ash was commonly consumed to prepare geopolymer concrete composites. The most important mechanical property for all types of concrete composites, including geopolymer concrete, is the compressive strength. However, in the structural design and construction field, the compressive strength of the concrete at 28 days is essential. Therefore, achieving an authoritative model for predicting the compressive strength of geopolymer concrete is necessary regarding saving time, energy, and cost-effectiveness. It gives guidance regarding scheduling the construction process and removal of formworks. In this study, Linear (LR), Non-Linear (NLR), and Multi-logistic (MLR) regression models were used to develop the predictive models for estimating the compressive strength of fly ash-based geopolymer concrete (FA-GPC). In this regard, a comprehensive dataset consists of 510 samples were collected in several academic research studies and analyzed to develop the models. In the modeling process, for the first time, twelve effective variable parameters on the compressive strength of the FA-GPC, including $SiO_2/Al_2O_3$ (*Si/Al*) of fly ash binder, alkaline liquid to binder ratio (*l/b*), fly ash (*FA*) content, fine aggregate (*F*) content, coarse aggregate (*C*) content, sodium hydroxide (*SH*)content, sodium silicate (*SS*) content, (*SS/SH*), molarity (*M*), curing temperature (*T*), curing duration inside ovens (*CD*) and specimen ages (*A*) were considered as the modeling input parameters. Various statistical assessments such as Root Mean Squared Error (RMSE), Mean Absolute Error (MAE), Scatter Index (SI), OBJ value, and the Coefficient of determination ($R^2$) were used to evaluate the efficiency of the developed models. The results indicated that the NLR model performed better for predicting the compressive strength of FA-GPC mixtures compared to the other models. Moreover, the sensitivity analysis demonstrated that the curing temperature, alkaline liquid to binder ratio, and sodium

**Competing interests:** The authors have declared that no competing interests exist.

silicate content are the most affecting parameter for estimating the compressive strength of the FA-GPC.

## 1. Introduction

It is commonly known that the production of Portland cement (PC) needs a considerable amount of energy as well as participates in about 7% of the total volume of carbon dioxide in the atmosphere. In the cement factories, around 50% of carbon dioxide is directly released into the air when the limestone heated in the calcination process, 40% delivers to the atmosphere as a result of the combustion of fuels to heat the rotary kiln, and the remaining 10% of the released carbon dioxide is measured for quarrying and transporting [1,2]. Also, around 2.8 tons of raw materials are needs for the manufacture of one ton of cement; this is a resource-exhausting process that consumes a large number of natural resources such as limestone and shale for the production of clinkers for cement [3]. Furthermore, approximately one trillion liters of mixing water are required to be used in the concrete industry annually [4]. In the same context, after the steel and aluminum industry, cement is one of the most energy-exhaustive construction materials that used around 110–120 kWh to produce one ton of cement in a typical cement plant alone [5].

Nevertheless, the majority of the cementing materials for the production of concrete are PC. Therefore, to decrease PC's environmental impact, a lot of research has been carried out to develop a new material to be an alternative to the PC [6]; geopolymer technology was developed first by Davidovits in France 1970 [7]. The green gas emission of geopolymer concrete (GC) is around 70% lower than the PC concrete due to the high consumption of waste materials in the mix proportions of the GC [8].

Geopolymers are one of the parts of mineral alumino-silicate polymers that generated from alkaline activation of different materials that rich in aluminosilicate materials, such as natural materials like metakaolin, by-product industrial materials like fly ash (FA), and the by-product of agro materials such as rice husk ash (RHA) [9]. The microstructure of geopolymer materials is amorphous and their chemical constituents are similar to the natural zeolitic materials. The mineral composition of the ash-based geopolymer and alkaline activators are the factors that affect the final product of the polymerization process. Also, the high temperature has usually accelerated the polymerization process [10,11]. So it can be concluded that geopolymer is the third generation of cementing materials after lime and cement [12]. Geopolymer concrete is a mixture of aluminosilicate binder, aggregates, alkaline solution, and water. Binder source materials such as FA, RHA, Ground Granulated Blast Furnace Slag (GGFBS), and metakaolin, or any hybridization between these ashes with or without PC. The most common industrial waste used as cementing material is FA, and it is divided into two classes: class F fly ash and Class C fly ash. The former FA has lower calcium content than class C FA [13]; FA has a variety of applications with high and low volumes for the production of different cementitious composites [14]. Aggregates consist of fine and coarse particles with required properties and gradation. The alkaline solution is a mixture of sodium hydroxide or potassium hydroxide with sodium silicate or potassium silicate. The polymerization between these ingredients produces a solid concrete almost like normal concrete [15].

The polymerization mechanism could be briefly explained as follows; in the first stage, dissolution of the silicate and aluminum elements of the binder inside the high alkalinity aqueous solution produces ions of silicon and aluminum oxide. In the second stage, a mixture of silicate, aluminate, and aluminosilicate species, which through a contemporaneous operation of

poly-condensation-gelation further condensation, finally produces an amorphous gel [16]. Several factors could influence the performance of GC such as type of binder, the concentration of the alkaline solution, the molarity of sodium hydroxide, sodium silicate to sodium hydroxide ratio, extra water, mix proportion, and curing method [17].

Compressive strength of all types of concrete composites, including GC is one of the most remarkable mechanical properties. Usually, it gives a general performance about the quality of the concrete composites [18]. The compressive strength test is conducted by following the standard test methods of ASTM C39 or BS EN 12390–3 [19,20]. In the literature, a variety of studies have been conducted to investigate the influence of several mixture parameters on the mechanical properties of FA-GPC, for instance, Hardjito et al., [21] studied the influence of different parameters such as molarity, sodium silicate to sodium hydroxide ratio, curing temperature, curing time, the dosage of high range water-reducing admixture, handling time, the water content in the mixture and age of concrete on the compressive strength of GC. They revealed that the higher molarity, sodium silicate to sodium hydroxide ratio, higher curing temperatures, and curing time gives higher compressive strength to the GC; At the same time, the increment in the percent of water leads to a reduction in the compressive strength [21].

On the other hand, a research study has been carried out by Patankar et al. [22] on the effect of water to geopolymer binder ratio on the performance of FA-GPC. They observed that the compressive strength was declined as the water to geopolymer binder ratio increased [22]. Similar observations can also found in other studies even though different mixture proportions were used [23].

One of the factors that affect the polymerization process is the type and quantity of the alkaline liquids by influencing the release of $Si^{4+}$ and $Al^{3+}$ from the base binders. Alkaline liquids of greater concentration are usually beneficial for getting higher compressive strength up to an optimal range [24]. Singhal et al. [25] prepared FA-GPC with different sodium hydroxide concentrations (molarity) range from 8 to 16 M. They observed that with the increment of the molarity of the geopolymer mixture compressive strength was increased. Also, sodium silicate ($Na_2SiO_3$) is a high viscosity solution that is generally used with sodium hydroxide (NaOH) to enhance the compressive strength of FA-GPC; $Na_2SiO_3$ helps the formation of geopolymer gels and gives a high compact microstructure to the final product of the FA-GPC [26]. Furthermore, a variety of ($Na_2SiO_3$/NaOH) ratio was used to prepare geopolymer concrete, for instance, a research study has been carried out by Topark-Ngarm et al., [27], who used a different ratio of $Na_2SiO_3$/NaOH, and they reported that with the increasing of $Na_2SiO_3$/NaOH, compressive strength was increased. In the same context, the amount of aggregate content in the geopolymer mixture proportions have influences on the compressive strength of the FA-GPC as investigated by Joseph and Mathew [28]. They performed an experimental laboratory work that used different aggregate volumes from 60% to 75%., and they concluded that the FA-GPC with the total aggregate content of 70%, the ratio of sand to the total aggregate of 0.35, the molarity of 10, *l/b* of 0.55, $Na_2SiO_3$/NaOH of 2.5, when cured for 1 day at 100˚C, provide the compressive strength of 52 MPa.

Another critical parameter that affects the performance of FA-GPC is the curing condition of the samples. Generally, there are various types of curing regimes, namely, ambient curing [29,30], heat curing [31,32], and steam curing [33–35]. Several types of research have been carried out on the mixed proportion of FA-GPC and its compressive strength when cured at temperatures varying from 23 to 120˚C. The polymerization process is rapidly increased with the increment of curing temperature which makes the GC gain up to 70% of its final strength when the specimens cured inside an oven at 65˚C for 24 hr. beyond which there is a peripheral enhance in the compressive strength after 28 days of maturity [36,37]. Further, heat curing regimes give higher compressive strength as compared to the ambient curing condition for the

same GC mixture [38–41]. Experimental program work was done by Joseph and Mathew [28]. They used different curing temperatures from 30 to 100˚C to cure their GC specimens; their results show that with the increment of curing temperature, the compressive strength was significantly increased. Similar results were obtained by Chithambaram et al. [42].

Achieving an authoritative model for predicting the compressive strength of GC is essential regarding saving in time, energy, and cost-effectiveness. It gives guidance about scheduling for the construction process and removal of framework elements [43]. The modeling of the compressive strength characteristic of the FA-GPC is essential regarding the possibility of changing or validating the GC mix proportions [44]. By selecting appropriate mixing proportions, economical and efficient designs will be accomplished. Therefore, a variety of researches have been tried to shorten the time of selecting an appropriate mix of proportions to get the targeted properties; among them is modeling with developing empirical equations. There are different ways for modeling the characteristics of construction materials, including statistical techniques, computational modeling, and nowadays developed techniques such as regression analysis [45,46]. A variety of factors affect the compressive strength of the FA-GPC; this leads to different compressive strength results; as a consequence, predicting compressive strength is a challenging task for researchers and engineers. Therefore, there is a need for numerical and mathematical models [47]. Machine learning's excellent ability regarding prioritization, optimization, forecasting, and planning was widely used in the various engineering fields [43]. In the literature, machine learning systems were used to model the various characteristics of different types of concrete composites such as compressive strength of green concrete [48], splitting tensile and flexural strength of recycled aggregate concrete [49], modulus of elasticity of recycled concrete aggregate [50,51], the compressive strength of high volume fly ash concrete [52], the compressive strength of eco-friendly GC containing natural zeolite and silica fume [53], splitting tensile strength of fiber-reinforced concrete [54], and so on.

In the literature, there is a lack of measuring effects of several mixture proportion parameters and different curing regimes on the compressive strength of FA-GPC from an early age to 112 days. Also, according to the comprehensive and systematic review on the FA-GPC, an authoritative and developed model which used a variety of parameters to predict the compressive strength of FA-GPC is very rare to be used by the construction industry. The majority of efforts have concerned a single scale model without covering broad laboratory work data or various parameters. Moreover, the compressive strength of FA-GPC is affected by more than one parameter; therefore, in this study, for the first time, in a single developed model, influences of twelve parameters, such as $SiO_2/Al_2O_3$ (*Si/Al*) of fly ash, alkaline liquid/binder (*l/b*), fly ash (*FA*) content, fine aggregate (*F*) content, coarse aggregate (*C*) content, sodium hydroxide (*SH*) content, sodium silicate (*SS*) content, (*SS/SH*) ratio, molarity (*M*), curing temperature (*T*), curing duration inside ovens (*CD*) and specimens ages (*A*) were investigated and quantified on the compressive strength of FA-GPC by using different model techniques, namely Linear Regression (LR), Nonlinear regression (NLR) and Multi-logistic Regression (MLR). They were used as predictive models for predicting the compressive strength of eco-efficient FA-GPC by using 510 samples from the literature studies.

## 2. Research significance

Provide multiscale models to predict the compressive strength of FA-GPC is the main scope of this study. Thus, a wide range of laboratory work data, about 510 tested specimens with various (Si/Al), (l/b), (FA), (F), (C), (SH), (SS), (SS/SH), (M), (T), (CD), and (A) were considered with different analysis approaches aiming: (i) to guarantee the construction industry to use the provided models without any theoretical; (ii) to carry out statistical analysis and recognize the

influence of various parameters on the compressive strength of FA-GPC; (iii) to quantify and provide a systematic multiscale model to predict the compressive strength of FA-GPC with the mixture propositions containing a various range of parameters; (iv) to discover the most authoritative model to predict the compressive strength of FA-GPC from three different model techniques (LR, NLR, and MLR) using statistical assessment tools.

## 3. Methodology

510 dataset was collected from past researches on FA-GPC. In the literature, there is a wide range of data regarding geopolymer concrete with different base source materials, including FA, GGBFS, RHA, silica fume (SF), Metakaolin (MK), red mud (RM), and so on. But in this paper, the authors take those papers that use fly ash (FA) as base source materials to prepare geopolymer concrete. The models used twelve input parameters to restrict authors from using more datasets in the developed models. The collected datasets were statistically analyzed and split into three groups. The larger group, which included 340 datasets, was used to create the models. The second group consists of 85 datasets used to test the proposed models, and the last group, which includes 85 datasets, was used to validate the provided models [43]. The dataset ranges can be seen in Table 1 that contains the range of all different parameters with the measured compressive strength of FA-GPC. The input dataset consists of the Si/Al range from 0.4–7.7, l/b range from 0.25–0.92, FA range from 254–670 kg/m$^3$, F range from 318–1196 kg/m$^3$, C range from 394–1591 kg/m$^3$, SH range from 25–135 kg/m$^3$, SS range from 48–342 kg/m$^3$, SS/SH range from 0.4–8.8, M range from 3–20, T range from 23–120˚C, CD range from 8–168 hr, and A range from 3–112 days. The former dataset was then used to propose different models to predict the compressive strength of FA-GPC, and compared with the actual experimental compressive strength (MPa); after that, the developed models were assessed by some statistical criteria such as coefficient of determination, root mean squared error, mean absolute error, scatter index and OBJ to indicate the most reliable and accurate model. Further details of the data collection and modeling work are summarized in the form of a flow chart, as depicted in Fig 1.

## 4. Statistical assessment

In the current section, a statistical analysis was carried out to see whether powerful relationships exist between input parameters and compressive strength of FA-GPC or not. In this regard, all considered dataset variables including (1) SiO$_2$/Al$_2$O$_3$ (*Si/Al*) of fly ash (2), alkaline liquid/binder (*l/b*) (3), fly ash content (*FA*) (4), fine aggregate content (*F*) (5), coarse aggregate content (*C*) (6), sodium hydroxide (*SH*) (7), sodium silicate (*SS*) (8), (*SS/SH*) ratio (9), molarity (*M*) (10), curing temperature (*T*) (11), curing duration inside ovens (*CD*) (12), specimens ages (*A*) was plotted and analyzed with compressive strength, also, the statistical criteria such as standard deviation, variance, skewness, and kurtosis were determined to illustrate the distribution of each variable with compressive strength. Regarding the kurtosis criteria, a high negative value demonstrates the shorter distribution tails compared to the normal distribution, while the longer tails represent the positive value. A high negative value indicates a long left tail for the skewness parameter, and a positive value represents a right tail. More information on each statistical criterion was reported by Sliva et al. [96]. Below sufficient information regarding each variable considered as the input parameter is present:

### a) SiO$_2$/Al$_2$O$_3$ (Si/Al)

Based on the dataset, which contains 510 data samples from literature, the Si/Al ratio of the fly ash was varied from 0.4 to 7.7 with an average of 2.7, the variance of 2.69, the standard deviation of 1.64, skewness of 2.5, and kurtosis of 5.03. Skewness belongs to distortion or asymmetry

**Table 1. Summary of different fly ash-based geopolymer concrete mixes.**

| Ref. | (Si/Al) | (l/b) | FA (kg/m³) | F (kg/m³) | C (kg/m³) | SH (kg/m³) | SS (kg/m³) | (SS/SH) | M | T (°C) | CD (hr.) | A (Day) | σc (Mpa) |
|---|---|---|---|---|---|---|---|---|---|---|---|---|---|
| [21] | 2 | 0.35 | 476 | 554 | 1294 | 48–120 | 48–120 | 0.4–2.5 | 8–14 | 24–90 | 8–96 | 3–94 | 17–64 |
| [22] | 4.3 | 0.35 | 334 | 555–632 | 1175–1329 | 58 | 58 | 1 | 13 | 90 | 8 | 7 | 17–61 |
| [23] | 2.2 | 0.3–0.45 | 400 | 830–895 | 830–895 | 32–52 | 85–129 | 2–3.3 | 12–18 | 50 | 48 | 7–28 | 16.36 |
| [25] | 2.1 | 0.45 | 350–400 | 505–533 | 1178–1243 | 45–52 | 112–129 | 2.5 | 8–16 | 24 | - | 3–28 | 7–41 |
| [27] | 2.2 | 0.5 | 414 | 588 | 1091 | 69–104 | 104–138 | 1–2 | 10–20 | 24–60 | 24 | 7–28 | 19–54 |
| [28] | 2.1 | 0.35–0.65 | 254–420 | 318–1198 | 394–1591 | 25–76 | 69–165 | 1.5–3.5 | 8–16 | 24–120 | 6–72 | 3–28 | 13–60 |
| [29] | 2.0 | 0.4 | 400 | 644 | 1197 | 53 | 107 | 2 | 10 | 24 | - | 3–56 | 5–23 |
| [30] | 1.8 | 0.4 | 394 | 554 | 1293 | 45 | 112 | 2.5 | 8 | 24 | - | 7–28 | 3–18 |
| [31] | 2.6 | 0.65 | 639 | 639 | 959 | 121 | 304 | 2.5 | 8_12 | 24 | - | 7–28 | 6–32 |
| [32] | 3.1 | 0.5 | 400 | 650 | 1206 | 50–70 | 140–154 | 2–2.75 | 14 | 60 | 168 | 7–28 | 30–36 |
| [33] | 1.6 | 0.35 | 408 | 647 | 1202 | 41 | 103 | 2.5 | 14 | 24–60 | 24 | 28 | 27–40 |
| [34] | 1.9 | 0.35 | 408 | 554 | 1294 | 41 | 103 | 2.5 | 8–14 | 60 | 24 | 7 | 40–64 |
| [35] | 1.9 | 0.35 | 356–444 | 554–647 | 1170–1248 | 36–44 | 89–111 | 2.5 | 14 | 60 | 24 | 7–28 | 24–63 |
| [38] | 2.1 | 0.38–0.46 | 350–400 | 540–575 | 1265–1343 | 38–53 | 95–132 | 2.5 | 16 | 24–90 | 24 | 3–28 | 2.6–44 |
| [39] | 0.4 | 0.4 | 350 | 650 | 1250 | 41 | 103 | 2.5 | 8 | 24–60 | 24 | 3–28 | 6–32 |
| [40] | 1.5 | 0.37 | 424 | 598 | 1169–1197 | 63 | 95 | 1.5 | 14 | 70 | 24 | 3–96 | 2–58 |
| [41] | 1.9 | 0.3 | 670 | 600 | 970 | 80 | 120 | 1.5 | 3–9 | 50 | 72 | 3–7 | 59–61 |
| [42] | 2.4 | 0.45 | 298–430 | 533–590 | 1243–1377 | 38–55 | 96–138 | 2.5 | 8–14 | 10–90 | 24 | 3–28 | 19–43 |
| [55] | 1.5–5.1 | 0.5–0.6 | 300–500 | 471–664 | 1000–1411 | 42–120 | 90–215 | 1.5–2 | 12–16 | 70 | 24 | 7 | 16–64 |
| [56] | 2.4 | 0.6 | 385 | 601.7 | 1203 | 66 | 165 | 2.5 | 12 | 80 | 24 | 3–28 | 74–81 |
| [57] | 1.8 | 0.45–0.55 | 300–350 | 698–753 | 1048–1131 | 38–55 | 96–118 | 2.5 | 10 | 100 | 24 | 7–28 | 26–36 |
| [58] | 3.0 | 0.81 | 409 | 686 | 909 | 129 | 204 | 1.58 | 15 | 80 | 24 | 28–96 | 22–27 |
| [59] | 2.3 | 0.4 | 394 | 646 | 1201 | 45 | 112 | 2.5 | 16 | 24–60 | 24 | 3–28 | 8–50 |
| [60] | 2.6 | 0.6 | 400 | 704 | 1056 | 68 | 171 | 2.5 | 10–16 | 60 | 24 | 7–28 | 25–32 |
| [61] | 1.5 | 0.35 | 408 | 554 | 1294 | 41 | 103 | 2.5 | 8 | 24 | - | 7–28 | 12–16 |
| [62] | 1.5 | 0.3–0.5 | 400–475 | 529–547 | 1235–1280 | 34–57 | 85–142 | 2.5 | 14 | 24 | - | 7–56 | 7–44 |
| [63] | 1.6 | 0.6 | 390 | 585 | 1092 | 67 | 167 | 2.5 | 8–18 | 24 | - | 28 | 23–32 |
| [64] | 2.1 | 0.35–0.38 | 408 | 660 | 1168–1201 | 41 | 103 | 2.5 | 10–16 | 24–50 | 24 | 28 | 25–72 |
| [65] | 2.8 | 0.55 | 356 | 554.4 | 1293 | 43–78 | 117–152 | 1.5–3.5 | 10 | 60 | 48 | 7–28 | 23–35 |
| [66] | 2.4–2.9 | 0.45 | 500 | 575 | 1150 | 64 | 160 | 2.5 | 14 | 24 | - | 28 | 44–52 |
| [67] | 2.4 | 0.4 | 440 | 723 | 1085 | 64 | 112 | 1.75 | 12 | 60 | 48 | 3–28 | 23–35 |
| [68] | 1.9 | 0.35 | 408 | 640–647 | 1190–1202 | 41 | 103 | 2.5 | 14–16 | 60 | 24 | 28 | 42–62 |
| [69] | 1.5–3.9 | 0.7–0.9 | 412–420 | 693–706 | 918–936 | 39–92 | 241–342 | 2.6–8.8 | 15 | 80 | 24 | 3–96 | 22–57 |
| [70] | 2.5 | 0.55 | 310 | 649 | 1204 | 48.86 | 122 | 2.5 | 10 | 80 | 24 | 28–96 | 44–47 |
| [71] | 1.9 | 0.4 | 400 | 651 | 1209 | 45 | 114 | 2.5 | 14 | 24 | - | 3–96 | 5–33 |
| [72] | 1.9 | 0.6 | 450 | 500 | 1150 | 135 | 135 | 1 | 10 | 40 | 24 | 7–96 | 18–49 |
| [73] | 1.7 | 0.4 | 400 | 554 | 1293 | 45 | 113 | 2.5 | 14 | 100 | 72 | 3–28 | 29–45 |
| [74] | 1.7 | 0.4 | 400 | 554 | 1293 | 45 | 113 | 2.5 | 14 | 100 | 72 | 3–28 | 29–45 |
| [75] | 1.9 | 0.37–0.4 | 408 | 647 | 1201 | 62–68 | 93–103 | 1.5 | 14 | 60 | 24 | 28 | 32–38 |
| [76] | 2.3–3.3 | 0.4 | 420–440 | 340–575 | 660–1127 | 60–68 | 150–169 | 2.5 | 12 | 80–120 | 72 | 7 | 21–61 |
| [77] | 3 | 0.35 | 409 | 549 | 1290 | 41 | 102 | 2.5 | 10 | 24 | - | 7–112 | 10–41 |
| [78] | 2.6–2.9 | 0.5 | 420 | 630 | 1090 | 60 | 150 | 2.5 | 12 | 80 | 24 | 7 | 32–41 |
| [79] | 2.3 | 0.5 | 368 | 554 | 1293 | 52 | 131 | 2.5 | 16 | 100 | 24 | 28 | 41 |
| [80] | 2.1–2.6 | 0.3 | 450 | 788–972 | 945–972 | 67 | 67 | 1 | 10 | 70 | 24 | 7–28 | 25–41 |
| [81] | 5.6 | 0.4 | 410 | 530 | 1044 | 67 | 117 | 1.74 | 10 | 24–75 | 26 | 7–180 | 4–36 |
| [82] | 2.3 | 0.45 | 500 | 550 | 1100 | 64.3 | 160.7 | 2.5 | 14 | 70 | 48 | 28 | 49.5 |
| [83] | 1.9 | 0.4 | 400 | 651–656 | 1209–1218 | 40–46 | 100–114 | 2.5 | 14 | 24 | _ | 28–90 | 25–41 |

*(Continued)*

**Table 1.** (Continued)

| Ref. | (Si/Al) | (l/b) | FA (kg/m³) | F (kg/m³) | C (kg/m³) | SH (kg/m³) | SS (kg/m³) | (SS/SH) | M | T (˚C) | CD (hr.) | A (Day) | σc (Mpa) |
|---|---|---|---|---|---|---|---|---|---|---|---|---|---|
| [84] | 1.7 | 0.4 | 400 | 554 | 1293 | 45 | 113 | 2.5 | 14 | 100 | 72 | 3–28 | 14–36 |
| [85] | 2.3 | 0.35–0.5 | 327–409 | 554–672 | 1201–1294 | 40–54 | 108–112 | 2–2.5 | 8–16 | 60 | 24 | 28 | 31–62 |
| [86] | 1.6 | 0.58 | 380 | 462 | 1386 | 62 | 156 | 2.5 | 10 | 60 | 24 | 28–56 | 18–23 |
| [87] | 1.9 | 0.4 | 394 | 554 | 1293 | 45 | 112 | 2.5 | 12 | 24–60 | 24 | 7–28 | 8–28 |
| [88] | 2.1 | 0.3–0.4 | 428 | 630 | 1170 | 44–57 | 114–122 | 2–2.5 | 8–14 | 60–90 | 24 | 3–7 | 20–49 |
| [89] | 1.5 | 0.3 | 563 | 732 | 5994 | 44 | 124 | 2.8 | 10 | 75 | 16 | 28 | 3345 |
| [90] | 7.7 | 0.4–0.6 | 345–394 | 554 | 1294 | 45–83 | 94–148 | 1.5–2.5 | 8–16 | 24 | - | 28 | 7–22 |
| [91] | 1.8 | 0.4 | 350 | 483 | 1081 | 40 | 100 | 2.5 | 14 | 24 | - | 7–28 | 3–23 |
| [92] | 1.7 | 0.45 | 436 | 654 | 1308 | 56 | 140 | 2.5 | 8 | 24 | - | 3–12 | 8–18 |
| [93] | 2.7 | 0.45 | 380 | 660 | 1189 | 48 | 122 | 2.5 | 8 | 24 | - | 28 | 30 |
| [94] | 1.6 | 0.35 | 500 | 623 | 1016 | 70 | 105 | 1.5 | 14–16 | 24 | - | 3–28 | 7–27 |
| [95] | 2.1 | 0.41 | 350 | 645 | 1200 | 41 | 103 | 2.5 | 8 | 24 | - | 3–56 | 7–21 |
| Remarks (Ranged are Varies Between) | 0.4–7.7 | 0.25–0.92 | 254–670 | 318–1196 | 394–1591 | 25–135 | 48–342 | 0.4–8.8 | 3–20 | 23–120 | 8–168 | 3–112 | 2–64 |

*(**Si/Al**) is a (SiO₂/Al₂O₃) ratio of fly ash, (**l/b**) is the alkaline liquid to binder ratio, (**FA**) is a fly ash content (kg/m³), (**F**) is a fine aggregate content (kg/m³), (**C**) is a coarse aggregate content (kg/m³), (**SH**) is a sodium hydroxide content (kg/m³), (**SS**) is a sodium silicate content (kg/m³), (**SS/SH**) is the ratio of sodium silicate to sodium hydroxide of the mix, (**M**) is the molarity (concentration of sodium hydroxide) of the mix, (**T**) is the curing temperature of the specimens and this is may be ambient curing or heat curing inside an oven (˚C), (**CD**) is the curing duration inside an oven (hr.), (**A**) is the age of samples at the time of testing (days) and (**σc**) is the measured compressive strength (MPa).

in a symmetrical normal distribution in a dataset. If the curve is moved to the right or the left side, it is stated to be skewed. Also, skewness could be quantified as an impersonation of the range to which a given distribution differs from a normal distribution. For instance, the skew of zero value was measured for normal distribution, while, right skew is an indication of log-normal distribution [97]. The variation between compressive strength and Si/Al, as well as the histogram analysis, is shown in Fig 2. As can be seen from figure a very poor relationship existed between compressive strength and the Si/Al ratio.

## b) Alkaline liquid/binder (l/b)

According to the dataset, which contains 510 data samples from past researches, the l/b ratio of the FBGC was varied from 0.25 to 0.92 with an average variance, standard deviation, skewness, and kurtosis of 0.5, 0.01, 0.1, 1.21, and 2.88, respectively, The variance informed of the degree of spread in dataset, the greater the spread of the data, the greater the variance is about the mean. The relationship between compressive strength and l/b with Histogram of FA-GPC mixtures is presented in Fig 3.

## c) Fly ash content (FA)

The content of fly ash in the mixture proportions of different FA-GPC for the collected data varied from 254 to 670 kg/m³. The FAs have different chemical compositions as well as various specific gravities ranging from 1.95 to 2.54. The average, standard deviation, variance, skewness, and kurtosis of the FA were 386 kg/m³, 63 kg/m³, 3974, 1.51, and 6.18. The kurtosis is a statistical indicator that explains how heavily the tails of a distribution of a set of data differ from the tails of the normal distribution. In addition, the kurtosis finds the heaviness of the distribution tails, while skewness measures the symmetry of the distribution. Moreover, the

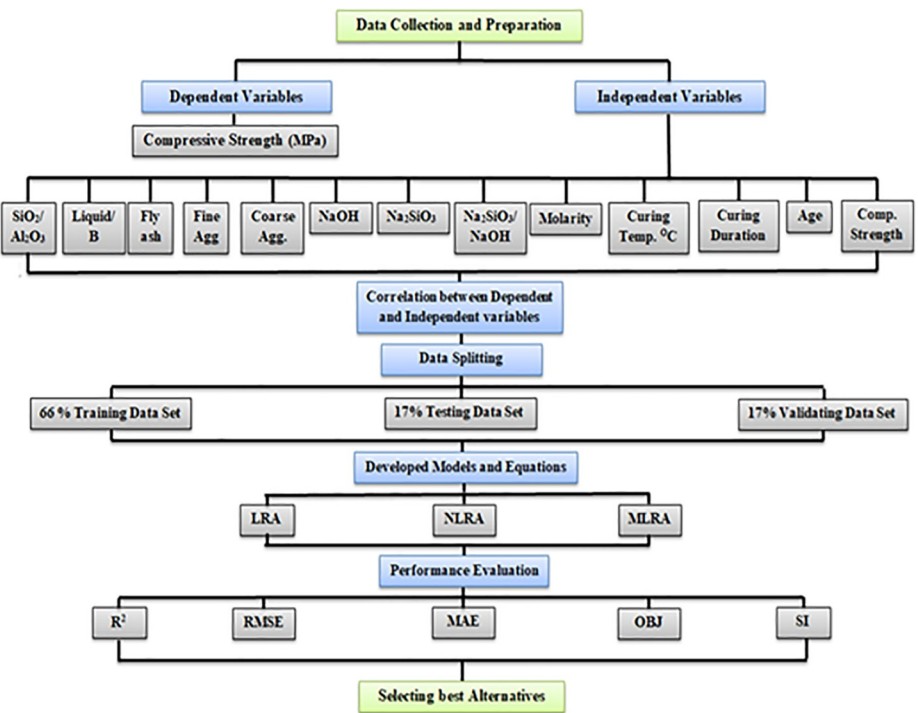

**Fig 1. The flow chart diagram process followed in this study.**

variation between compressive strength and FA content and Histogram of FA-GPC mixtures is reported in Fig 4.

### d) Fine aggregate content (*F*)

In the past studies, the fine aggregate was a river and crushed sand with a maximum aggregate size of 4.75 mm, and specific gravity ranged between 2.60–2.75. Also, its gradation satisfied the limitations of ASTM C 33. Fine aggregate content for the collected 510 datasets was varying from 318 to 1196 kg/m$^3$ for the mixtures of FA-GPC, and it has an average of 615kg/m$^3$, a standard deviation of 100 kg/m$^3$, a variance of 10047. Other statistical variables for the fine aggregate content in the FA-GPC mixtures, such as skewness and kurtosis, are 1.75 and 5.56. The relationship between compressive strength and fine aggregate content with a Histogram of FA-GPC mixtures is illustrated in Fig 5.

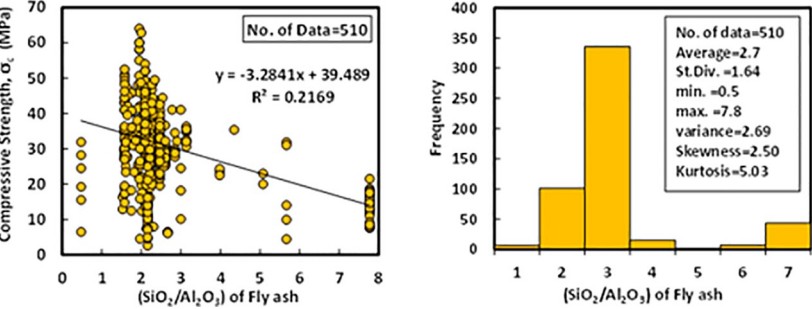

**Fig 2. Variation between compressive strength and (SiO$_2$/Al$_2$O$_3$) ratio of fly ash with the histogram of fly ash-based geopolymer concrete mixtures.**

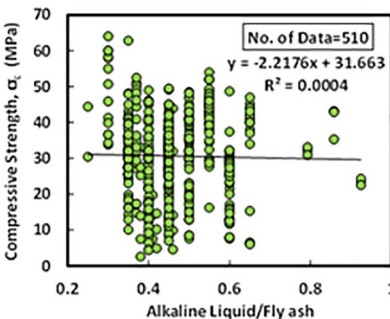
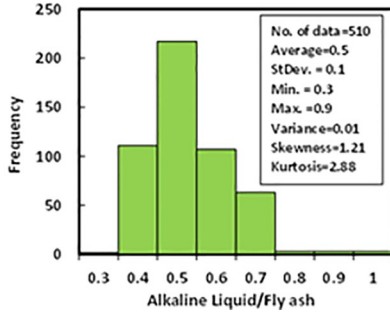

**Fig 3. Variation between compressive strength and (alkaline liquid/fly ash) ratio with a histogram of fly ash-based geopolymer concrete mixtures.**

### e) Coarse aggregate content (*C*)

The crushed stone or gravel with a maximum aggregate size of 20 mm was used in the literature as coarse aggregate for the production of FA-GPC. Based on the collected 510 dataset from different FA-GPC mixture proportions, coarse aggregate content varied between 394 to 1591 kg/m$^3$. The statistical analysis of the dataset shows that the average of the coarse aggregate content was 1187 kg/m$^3$, the standard deviation was 146.8 kg/m$^3$, the variance was 21557, the skewness was -1.69, and the kurtosis was 4.5. Variation between compressive strength and coarse aggregate content with Histogram of FA-GPC mixtures are presented in Fig 6.

### f) Sodium hydroxide (*SH*)

The content of the sodium hydroxide (NaOH) for the collected 510 datasets varied from 25 to 135 kg/m$^3$, with an average of 54.3 kg/m$^3$, the standard deviation of 16.11 kg/m$^3$, and a variance of 259. The skewness and kurtosis were 1.69 and 4.55, respectively. The purity of the SH was above 97% of all the FA-GPC mixtures, and pellets and flakes were the two main states of the SH in all the mixtures. The relationship between compressive strength and sodium hydroxide with a Histogram of FA-GPC mixtures are illustrated in Fig 7.

### g) Sodium silicate (*SS*)

Based on the dataset, which contains 510 data samples from literature, the content of SS was varied between 48 to 342 kg/m$^3$. The constituents of the SS were $SiO_2$, $Na_2O$, and water. The range of $SiO_2$ was varying from 28 to 37%, $Na_2O$ was in the range of 8 to 18%, and the percent of water in the SS was in the range of 45 to 64%. The statistical analysis for the collected data of

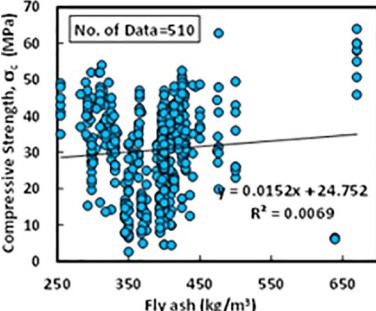
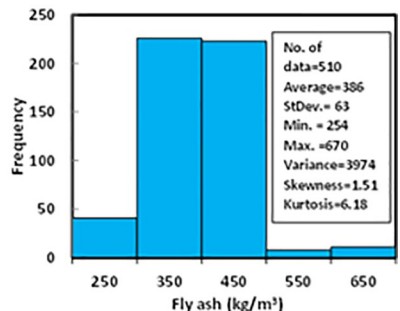

**Fig 4. Variation between compressive strength and fly ash content with a histogram of fly ash-based geopolymer concrete mixtures.**

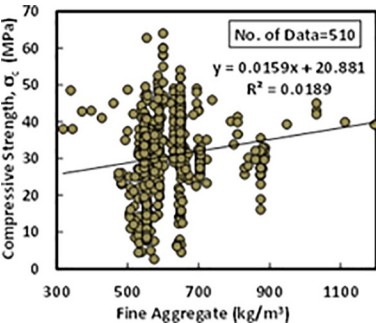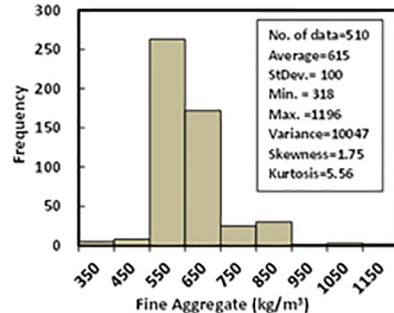

**Fig 5. Variation between compressive strength and fine aggregate content with the histogram of fly ash-based geopolymer concrete mixtures.**

SS revealed that the average content of SS in the FA-GPC was 123.4 kg/m$^3$, the standard deviation was 36.2 kg/m$^3$, the variance was 1313, skewness was 2.89, and kurtosis was 12.8. Variation between compressive strength and sodium silicate ($Na_2SiO_3$) content with Histogram of FA-GPC mixtures are presented in Fig 8.

### h) *SS/SH*

Referring to the collected data, the ratio of $Na_2SiO_3$ to NaOH was varied from 0.4 to 8.8, with an average of 2.4. The standard deviation, variance, skewness, and kurtosis were 0.68, 0.47, 4.71, and 45.9, respectively. The relationship between compressive strength and SS/SH with Histogram of FA-GPC mixtures is shown in Fig 9.

### i) Molarity (*M*)

According to the dataset, which contains 510 data samples from literature, the sodium hydroxide concentration (molarity) was varying from 3 to 20 M, with an average of 11.9 M, the standard deviation of 2.8 M, the variance of 7.83, the skewness of 0.14 and the kurtosis of -0.41. Variation between compressive strength and molarity with Histogram of FA-GPC mixtures are illustrated in Fig 10.

### j) Curing temperature (T)

The statistical analysis for the total collected data of the 510 dataset shows that the range of the curing temperature was varied from 23 to 120˚C, with an average of 58.6˚C and standard

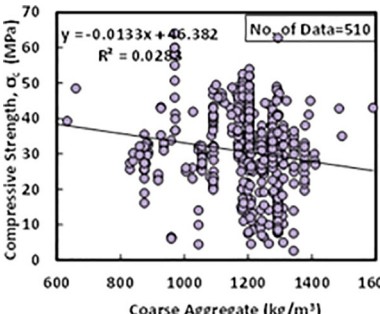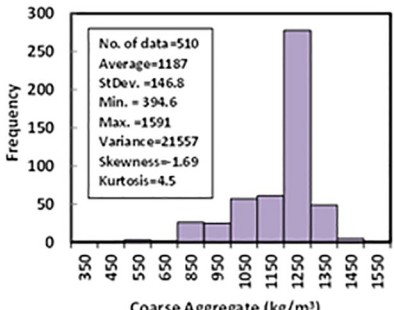

**Fig 6. Variation between compressive strength and coarse aggregate content with a histogram of fly ash-based geopolymer concrete mixtures.**

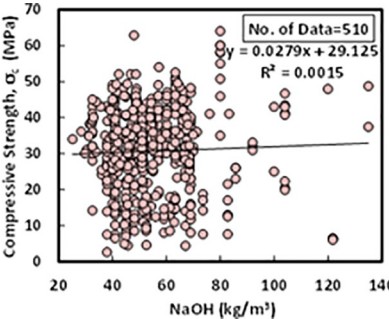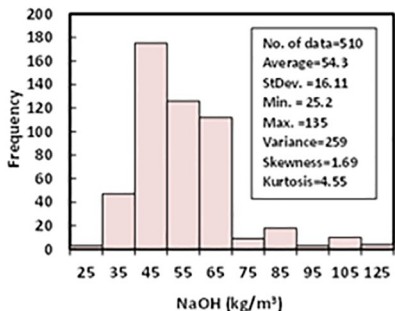

**Fig 7. Variation between compressive strength and sodium hydroxide content with a histogram of fly ash-based geopolymer concrete mixtures.**

deviations of 27.9˚C. Besides, the variance, skewness, and kurtosis were 7, 0.05, and -1.16, correspondingly. The relationship between compressive strength and curing temperature with a Histogram of FA-GPC mixtures is shown in Fig 11.

### k) Oven curing duration (*CD*)

The duration of heating samples in the oven with the selected temperatures was another independent variable that is collected from the past different research studies. The statistical analysis revealed that the minimum curing duration of the collected data set was 8 hr. The maximum CD inside ovens was 168 hr. Moreover, the average of CD was measured as 29 hr. the other statistical indications such as standard deviation, variance, skewness, and kurtosis were recorded as 19.86 hr, 395, 5.66, and 35.6, respectively. Variation between compressive strength and the oven curing duration with Histogram of FA-GPC mixtures are illustrated in Fig 12.

### l) Specimens ages (*A*)

Another independent variable collected in the literature papers is the age of FA-GPC specimens. The collected data contain the ages of the samples range from 3 up to 112 days. Other statistical measuring devices such as standard deviation, variance, skewness, and kurtosis were calculated as 15.65 days, 245, 2.67, and 10.75, correspondingly. Variation between compressive strength and specimens ages with Histogram of FA-GPC mixtures are shown in Fig 13.

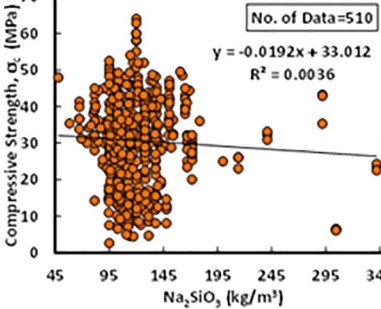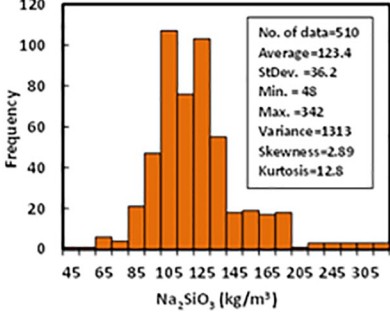

**Fig 8. Variation between compressive strength and sodium silicate content with histogram of fly ash-based geopolymer concrete mixtures.**

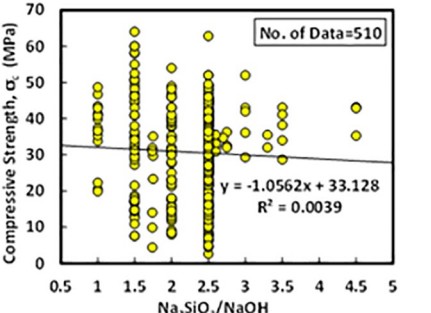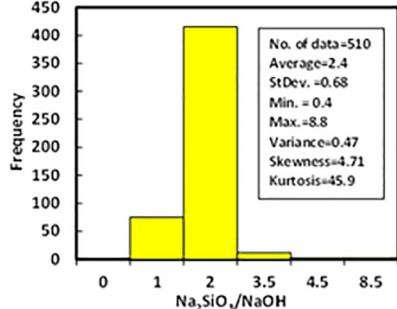

**Fig 9. Variation between compressive strength and (SS/SH) ratio with a histogram of fly ash-based geopolymer concrete mixtures.**

## m) Compressive strength (σc)

The measured compressive strength of the 510 collected data from the literature studies was shown in Table 1; the compressive strength of the FA-GPC was in the range of 2 to 64 MPa, with an average of 30.6 MPa. The statistical analysis for the other dataset distribution indications such as standard deviation, variance, skewness, and kurtosis was 11.6 MPa, 133.8, -0.16, and -0.3, respectively.

## 5. Modeling

Based on the coefficient of determination ($R^2$) and statistical analysis, there are no direct relationships between the compressive strength and the constituents of the FA-GPC at different curing regimes as shown in Figs 2–13. Therefore, three different models, as reported below, are proposed to evaluate the impact of different mixture proportions mentioned above on the compressive strength of FA-GPC.

The models proposed in this study are used to predict the compressive strength of FA-GPC and select the best model, which gives a better estimation of compressive strength compared with the measured compressive strength from the experimental data. All the collected datasets were randomly split in to three parts, namely training, testing, and validating datasets [43]. 340 Training dataset is used to train the LR, NLR, and MLR model and obtain the optimal weights and biases, while 85 testing dataset is used to confirm the fulfillment of the proposed models. Moreover, 85 validating datasets are used to explore the generality of the models and prohibition of the over-fitting problem in the case of classical training algorithms. The comparison

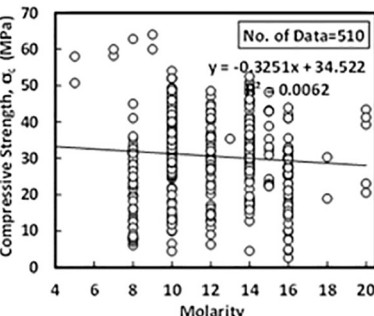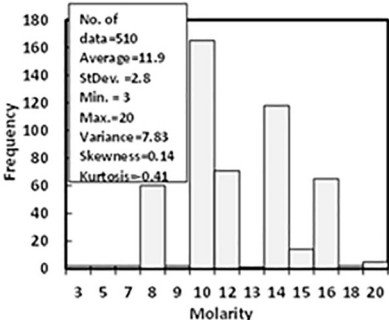

**Fig 10. Variation between compressive strength and molarity with a histogram of fly ash-based geopolymer concrete mixtures.**

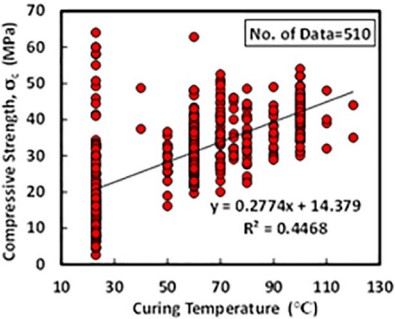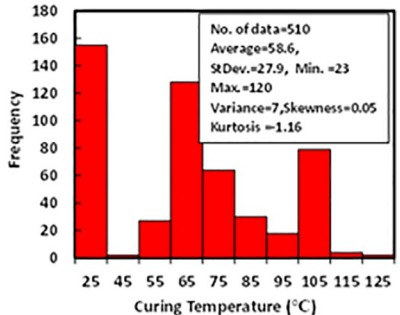

**Fig 11. Variation between compressive strength and curing temperature histogram of fly ash-based geopolymer concrete mixtures.**

among model predictions was made based on the following assessment criteria: the model should be scientifically valid, it should give less percentage of error between the measured and predicted data, lower RMSE, OBJ, SI, and higher $R^2$ value.

## a) Linear regression model (LR)

One of the most common methods to predict the compressive strength of concrete is the linear regression model (LR) [98], as shown in Eq 1, and it is considered as a general form of linear regression model [52,97]

$$\sigma c = a + b(l/b) \tag{1}$$

Where, $\sigma c$, $l/b$, $a$ $and$ $b$ represents compressive strength, liquid to binder ratio and equation parameters, respectively. However, other components of FA-GPC mixtures that influence the compression strength, such as curing regime and time and different mix proportions, are not included in the equation above. Therefore, to have more reliable and scientific observations, Eq 2 is proposed to include all other mix proportions and variables that may impact the compressive strength of FA-GPC.

$$\sigma c = a + b\left(\frac{Si}{Al}\right) + c\left(\frac{l}{b}\right) + d(FA) + e(F) + f(C) + g(SH) + h(SS) + i\left(\frac{SS}{SH}\right) + j(M)$$
$$+ k(A) + l(T) + m(CD) \tag{2}$$

Where: ($Si/Al$) is the ratio of $SiO_2$ to $Al_2O_3$ of the fly ash, ($l/b$) is the alkaline liquid to the

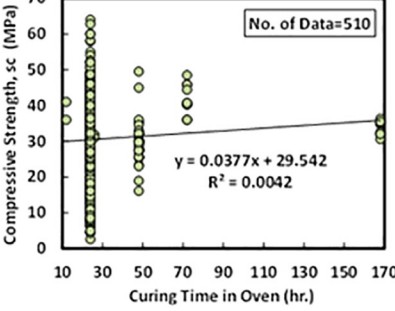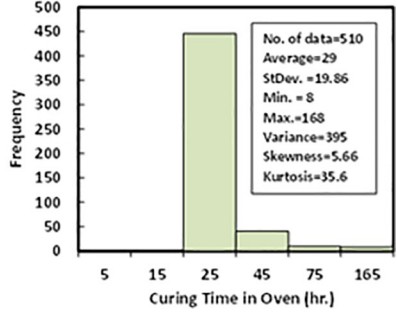

**Fig 12. Variation between compressive strength and curing duration with histogram of fly ash-based geopolymer concrete mixtures.**

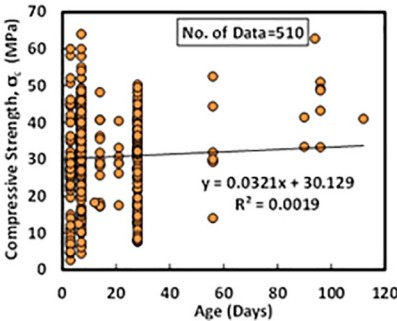 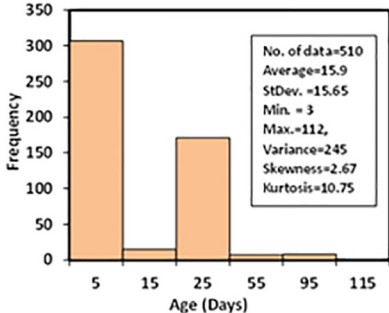

**Fig 13. Variation between compressive strength and curing duration histogram of fly ash-based geopolymer concrete mixtures.**

binder ratio, ($FA$) is the fly ash content (kg/m$^3$), ($F$) is the fine aggregate content (kg/m$^3$), ($C$) is the coarse aggregate content (kg/m$^3$), ($SH$) is the sodium hydroxide content (kg/m$^3$), ($SS$) is the sodium silicate content (kg/m$^3$), ($SS/SH$) is the ratio of sodium silicate to the sodium hydroxide, ($M$) is the sodium hydroxide concentration (Molarity), ($T$) is the curing temperature (˚C), ($CD$) is the curing duration inside ovens (hr) and ($A$) is the ages of the specimens (days). While $a, b, c, d, e, f, g, h, i, j, k, l,$ and $m$ are the model parameters. This developed equation is a unique equation that involves a wide range of independent variables to produce FA-GPC that may be very useful for the construction industry. The proposed Eq 2 can be considered as an extent for Eq 1 since all variables can be adapted linearly.

## b) Nonlinear regression model (NLR)

To propose a NLR model, Eq 3 could be considered as a general form [99,100]. The interrelation between different variables in Eqs 1 and 2 can be represented in Eq 3 to predict the compression strength of FA-GPC mixtures.

$$\sigma c = a * \left(\frac{Si}{Al}\right)^b * \left(\frac{l}{b}\right)^c * (FA)^d * (F)^e * (C)^f * (SH)^g * (SS)^h * \left(\frac{SS}{SH}\right)^i * (M)^j * (A)^k + l$$

$$* \left(\frac{Si}{Al}\right)^m * \left(\frac{l}{b}\right)^n * (FA)^o * (F)^p * (C)^q * (SH)^r * (SS)^s * \left(\frac{SS}{SH}\right)^t * (M)^u * (A)^v * (T)^w$$

$$* (CD)^x \tag{3}$$

Where: ($Si/Al$) is the ratio of SiO$_2$ to Al$_2$O$_3$ of the fly ash, ($l/b$) is the alkaline liquid to the binder ratio, ($FA$) is the fly ash content (kg/m$^3$), ($F$) is the fine aggregate content (kg/m$^3$), ($C$) is the coarse aggregate content (kg/m$^3$), ($SH$) is the sodium hydroxide content (kg/m$^3$), ($SS$) is the sodium silicate content (kg/m$^3$), ($SS/SH$) is the ratio of sodium silicate to the sodium hydroxide, ($M$) is the sodium hydroxide concentration (Molarity), ($T$) is the curing temperature (˚C), ($CD$) is the curing duration inside ovens (hr.) and ($A$) is the ages of the specimens (Days). While, $a, b, c, d, e, f, g, h, i, j, k, l, m, n, o, p, q, r, s, t, u, v, w,$ and $x$ are the model parameters.

## c) Multi-logistic regression model (MLR)

Same as the former models, multi-logistic regression analysis model was carried out for the collected datasets, and the general form of the MLR is shown in Eq 4 based on the research studied that had been conducted by Mohammed et al. [51]. MLR is used to clarify the

difference between a nominal predictor variable and one or more independent variables.

$\sigma_c = a47208$ *een model predictions of compressive strength of fly ash based geopolymer concrete mixtures using training data*161616

$$* \left(\frac{Si}{Al}\right)^b * \left(\frac{l}{b}\right)^c * (FA)^d * (F)^e * (C)^f * (SH)^g * (SS)^h * \left(\frac{SS}{SH}\right)^i * (M)^j * (A)^k * (T)^l * (CD)^m \tag{4}$$

Where: ($Si/Al$) is the ratio of $SiO_2$ to $Al_2O_3$ of the fly ash, ($l/b$) is the alkaline liquid to the binder ratio, ($FA$) is the fly ash content (kg/m$^3$), ($F$) is the fine aggregate content (kg/m$^3$), ($C$) is the coarse aggregate content (kg/m$^3$), ($SH$) is the sodium hydroxide content (kg/m$^3$), ($SS$) is the sodium silicate content (kg/m$^3$), ($SS/SH$) is the ratio of sodium silicate to the sodium hydroxide, ($M$) is the sodium hydroxide concentration (Molarity), ($T$) is the curing temperature (°C), ($CD$) is the curing duration inside ovens (hr.) and ($A$) is the ages of the specimens (Days). While $a$, $b$, $c$, $d$, $e$, $f$, $g$, $h$, $i$, $j$, $k$, $l$, and $m$ are the model parameters.

## 6. Model performance assessment criteria

In order to evaluate and assess the efficiency of the proposed models, various performance parameters, including the coefficient of determination ($R^2$), Root Mean Squared Error (RMSE), Mean Absolute Error (MAE), Scatter Index (SI), and OBJ, were used, which are defined as follows:

$$R^2 = \left(\frac{\sum_{p=1}^{p}(t_p - t')(y_p - y')}{\sqrt{[\sum_{p=1}^{p}(t_p - t')^2][\sum_{p=1}^{p}(y_p - y')^2]}}\right)^2 \tag{5}$$

$$RMSE = \sqrt{\frac{\sum_{p=1}^{p}(y_p - t_p)^2}{p}} \tag{6}$$

$$MAE = \frac{\sum_{p=1}^{p}|(y_p - t_p)|}{p} \tag{7}$$

$$SI = \frac{RMSE}{t'} \tag{8}$$

$$OBJ = \left(\frac{n_{tr}}{n_{all}} * \frac{RMSE_{tr} + MAE_{tr}}{R_{tr}^2 + 1}\right) + \left(\frac{n_{tst}}{n_{all}} * \frac{RMSE_{tst} + MAE_{tst}}{R_{tst}^2 + 1}\right) + \left(\frac{n_{val}}{n_{all}} * \frac{RMSE_{val} + MAE_{val}}{R_{val}^2 + 1}\right) \tag{9}$$

Where: $y_p$ and $t_p$ are the predicted and the measured values of the pth pattern, correspondingly, and $t'$ and $y'$ are the averages of the measured and the predicted values, respectively. $tr$, $tst$, and $val$ are referred to as training, testing, and validating datasets, respectively and $n$ is the number of patterns (collected data) in the corresponding dataset.

Except for the $R^2$ value, the best value for other assessment parameters is zero. However, the best value for $R^2$ is one. Regarding the SI parameter, it can be said that a model has a poor performance when SI > 0.3, a fair performance when 0.2 < SI < 0.3, a good performance when 0.1 < SI < 0.2, and an excellent performance when SI < 0.1 [43,101]. Moreover, in Eq (9) the OBJ parameter was also used to assess the efficiency of the proposed models as an integrated performance parameter.

## 7. Analysis and outputs

### a) LR model

The comparison between predicted and measured compressive strengths of FA-GPC for training, testing and validating datasets are presented in Fig 14A–14C, respectively. The model parameters observed that the l/b ratio and the ratio of sodium silicate to the sodium hydroxide significantly affects the compressive strength of FA-GPC. For the current model the weight of each parameter on the compressive strength of FA-GPC was determined by optimizing the sum of error squares and the least square method, which implemented in Excel program using Solver to calculate the ideal value (a specific value, minimum or maximum) for the equation in one cell named the objective cell. This object cell was subject to certain limits or constraints on the values of other equation cells in the worksheet [52]. Based on the linear regression analysis model, it was observed that, among the whole model input parameters, the ratio of alkaline liquid to the binder ration (l/b) and the sodium silicate to the sodium hydroxide ratio of the GC mixture have a great influence on the compressive strength of the FA-GPC which it is matched with the experimental results presented in the literature [21,23,25,28,55]. The equation for the LR model with different weight parameters can be written as follows as reported in Eq 10.

$$\sigma c = -66.8 - 1.697\left(\frac{Si}{Al}\right) + 187.75\left(\frac{l}{b}\right) + 0.246(FA) - 0.016(F) - 0.012(C) - 0.334(SH)$$
$$- 0.538(SS) + 0.942\left(\frac{SS}{SH}\right) + 0.179(M) + 0.228(A) + 0.342(T) + 0.01(CD) \qquad (10)$$

The studied datasets have a ±20% error line for the training data and -15% and +20% error lines for both testing and validating datasets. Nevertheless, the developed model slightly overestimated the low strength FA-GPC mixes and underestimated the high strength FA-GPC. Also, the residual compressive strength between the predicted and measured compressive strength for the LR model by using training, testing, and validating dataset were compared, as shown in Fig 15. This model's evaluation parameters, such as $R^2$, RMSE, and MAE are 0.8369, 4.65 MPa, and 3.76 MPa, respectively. Moreover, as reported from Figs 16 and 17, the OBJ and SI values for the current model are 3.09 and 0.15 for the training dataset.

### b) NLR model

The relationships between the predicted compressive strength and measured compressive strength obtained from experimental programs of FA-GPC mixtures for training, testing, and

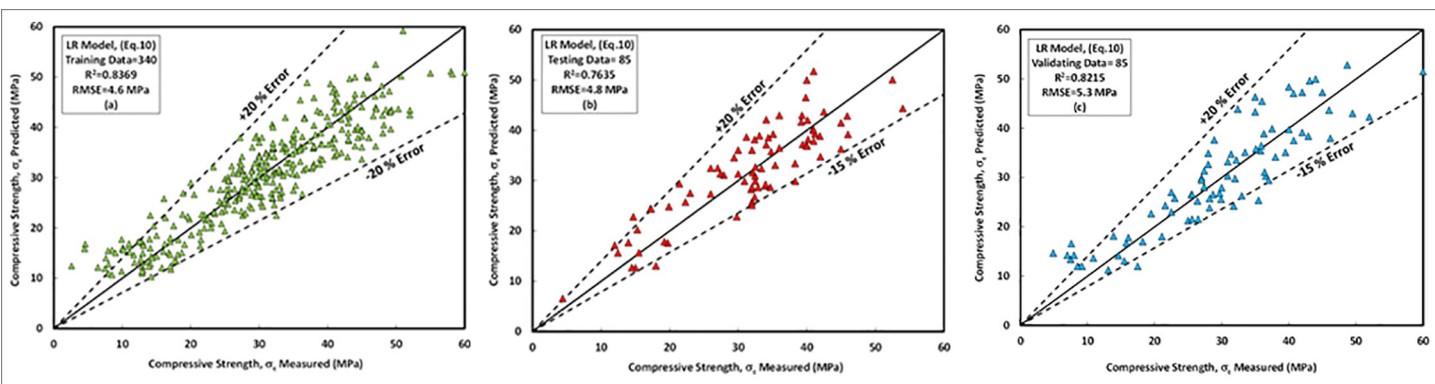

**Fig 14.** Comparison between measured and predicted compressive strength of fly ash-based geopolymer concrete mixture using LR model, (a) training data, (b) testing data, (c) validating data.

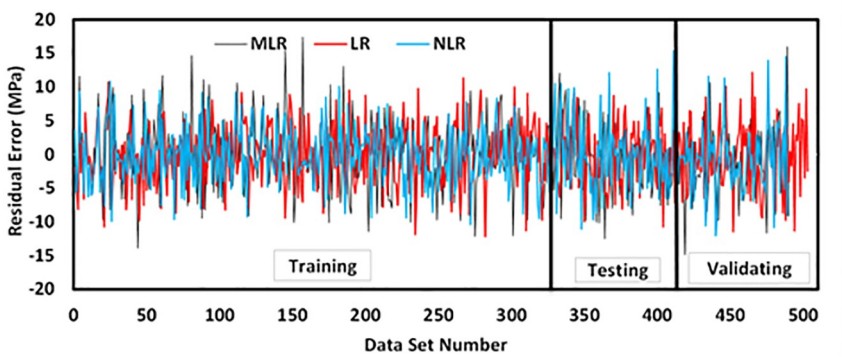

**Fig 15. Residual error diagram of compressive strength of fly ash-based geopolymer concrete mixtures using training, testing, and validating dataset for all models.**

validating datasets are presented in Fig 18A–18C, respectively. The most important parameters which affects the compressive strength of FA-GPC mixtures according to this model are the curing temperature and sodium silicate content. This was also approved by several experimental programs from past studies, in which increasing the sodium silicate content and increasing the curing temperature was resulted in the increasing the compressive strength of FA-GPC mixtures significantly [21,27,31,38,40,55,81,87,92]—the proposed equation for NLR model with different variable parameters presented in Eq 11.

$$
\begin{aligned}
\sigma_C = &-1997208 * \left(\frac{Si}{Al}\right)^{-0.508} * \left(\frac{l}{b}\right)^{-1.606} * (FA)^{-2.134} * (F)^{0.016} * (C)^{0.089} * (SH)^{-0.27} * (SS)^{0.274} \\
&* \left(\frac{SS}{SH}\right)^{-0.533} * (M)^{0.117} * (A)^{-0.305} + 9993.13 * \left(\frac{Si}{Al}\right)^{-0.423} * \left(\frac{l}{b}\right)^{-0.068} * (FA)^{-0.368} \\
&* (F)^{-0.151} * (C)^{-0.184} * (SH)^{-0.426} * (SS)^{-0.0007} * \left(\frac{SS}{SH}\right)^{-0.453} * (M)^{0.134} * (A)^{-0.022} \\
&* (T)^{0.352} * (CD)^{-0.064}
\end{aligned}
\tag{11}
$$

The studied datasets have a ±20% error line for the training data and -15% and +20% error lines for both testing and validating datasets. Similar to the LR model, this model slightly

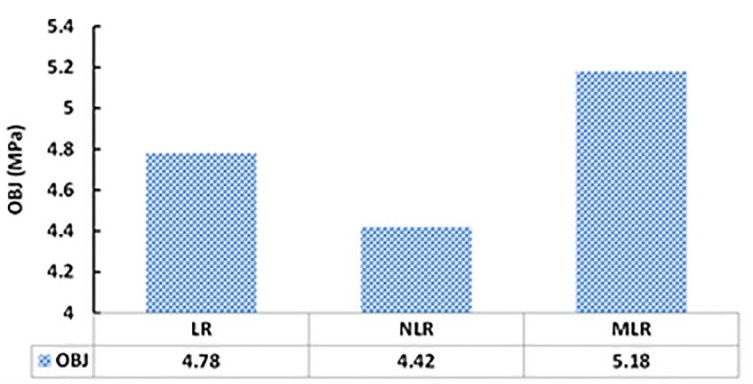

**Fig 16. The OBJ values of all developed models.**

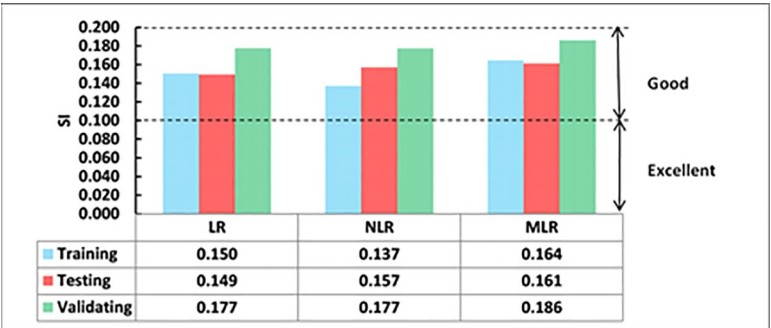

**Fig 17. Comparing the SI performance parameter of different developed models.**

underestimated the high strength FA-GPC mixes and overestimated the low strength FA-GPC. Also, the residual compressive strength was shown in Fig 15, which shows the residual error between the predicted and measured compressive strength for the NLR model by using training, testing, and validating datasets. In addition, the assessment parameters for this model, such as $R^2$, RMSE, and MAE, are 0.8576, 4.19 MPa, and 3.35 MPa, respectively, and the other assessment tools such as OBJ and SI are 2.71 and 0.14 correspondingly, as illustrated from Figs 16 and 17.

### c) MLR model

The proposed equation for MLR model with different variable parameters presented in Eq 12. In the MLR model, like other developed models, the curing temperature, sodium silicate content, an alkaline liquid to the binder ratio were the most significant independent variables that affect on the compressive strength of the FA-GPC that is matched with the experimental works presented in the literature [21,23,25,27,28,31,38,40,55,81,87,92]. The relationships between the predicted and measured compressive strength of the training data set for FA-GPC was shown in Fig 19A. Further, same as the two previous models, this model was checked by two sets of data (testing and validating dataset) to show their efficiency for other data out of the model data (training data); the results show that this model can be used to predict the compressive strength of FA-GPC just by substitute the independent variables into the developed

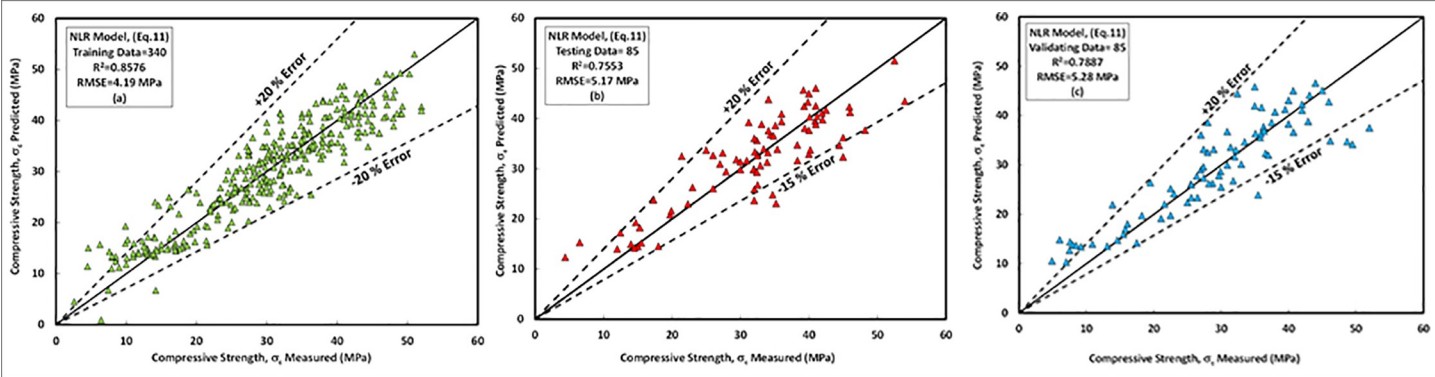

**Fig 18.** Comparison between measured and predicted compressive strength of fly ash-based geopolymer concrete mixture using NLR model, (a) training data, (b) testing data, (c) validating data.

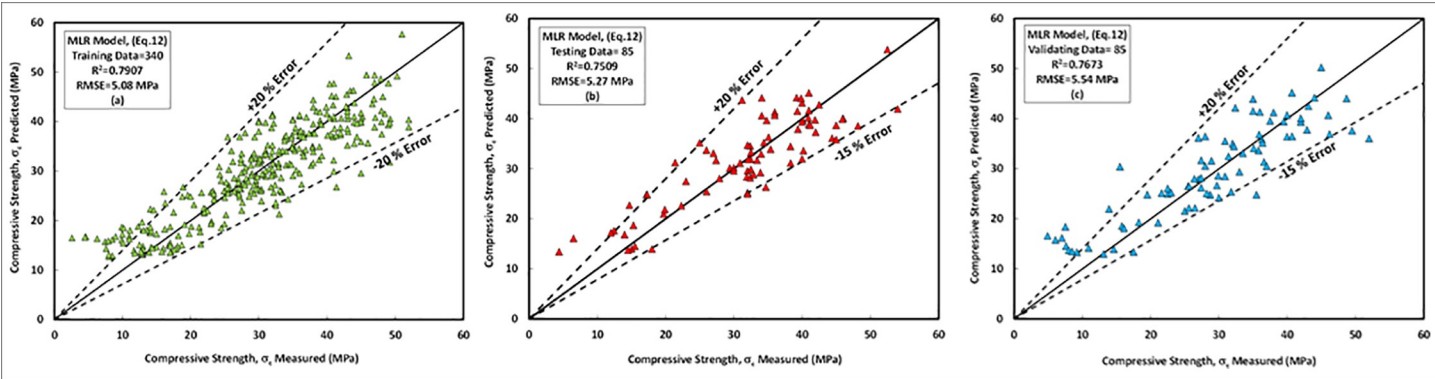

**Fig 19.** Comparison between measured and predicted compressive strength of fly ash-based geopolymer concrete mixture using MLR model, (a) training data, (b) testing data, (c) validating data.

equation as shown in Fig 19B and 19C.

$\sigma_c = 147.1447208$ *een model predictions of compressive strength of fly ash based geopolymer concrete mixtures using training data*191919

$$* \left(\frac{Si}{Al}\right)^{-0.383} * \left(\frac{l}{b}\right)^{0.350} * (FA)^{0.195} * (F)^{-0.212} * (C)^{-0.236} * (SH)^{-0.715} * (SS)^{0.393} * \left(\frac{SS}{SH}\right)^{-0.81} * (M)^{0.086} * (A)^{0.128} * (T)^{0.534} * (CD)^{-0.046} \qquad (12)$$

Similar to other models, the studied datasets have a ±20% error line for the training data and -15% and +20% error lines for both testing and validating datasets, which indicated that almost all checked results were in ± 20% error lines. Finally, the residual compressive strength for the MLRA model was shown in Fig 15 for the predicted and measured compressive strength using training, testing, and validating datasets. Furthermore, the assessment criteria for this model, such as $R^2$, RMSE, MAE, OBJ, and SI are 0.7907, 5.08 MPa, 3.95 MPa, 3.4, and 0.17, respectively, for the training dataset.

## 8. Comparison between developed models

As mentioned previously, five different statistical tools, which are RMSE, MAE, SI, OBJ, and $R^2$ was used to evaluate the efficiency of the developed models. Among the three different models, the NLR model has higher $R^2$ with lower RMSE and MAE values compared to LR and MLR models. Also, Fig 20 presents the comparison between model predictions of the compressive strength of FA-GPC mixtures using training data. Moreover, Fig 15 shows the residual error for all models using training, testing, and validating datasets. It can be noticed from both figures that the predicted and measured values of compressive strength are closer for the NLR mode, which indicates the superior performance of the NLR model compared to other models.

The OBJ values for all proposed models are given in Fig 16. The values for LR, NLR, and MLR are 4.78, 4.42, and 5.18, respectively. The OBJ value of the NLR model is 8.1% less than the LR model and 17.2% lower than the NLR models. This also demonstrates that the NLR model is more efficient for predicting the compressive strength of FA-GPC mixtures.

The values of the SI assessment parameter for the proposed models in the training, validating, and testing phases are presented in Fig 17. As can be seen from Fig 17, for all models and all phases (Training, testing, and validating), the SI values were between 0.1 and 0.2, indicating good performance for all models. However, similar to the other performance parameters the NLR model has lower SI values compared to other models. The NLR model has 9.4% and 19.7% lower SI values than LR and MLR models, correspondingly. This also illustrated that the

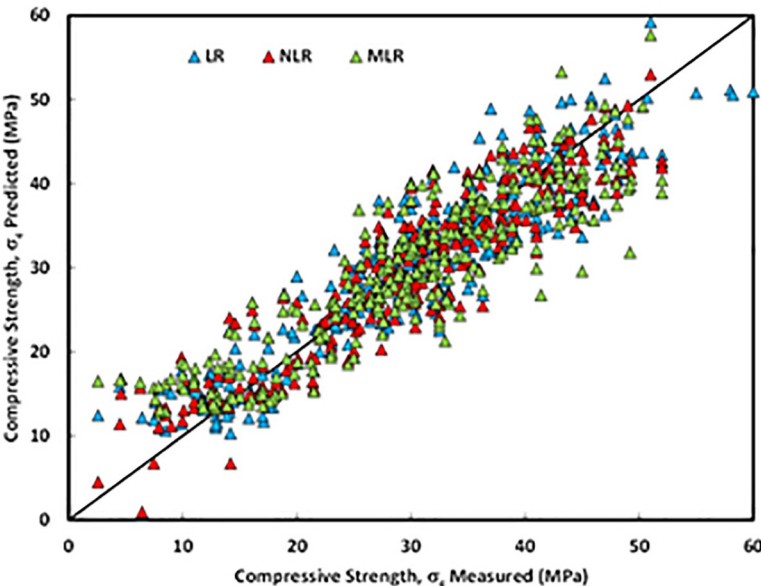

**Fig 20. Compression between model predictions of compressive strength of fly ash-based geopolymer concrete mixtures using training data.**

NLR model is more efficient and performed better compared to LR and MLR models for predicting the compressive strength of FA-GPC.

## 9. Sensitivity investigation

In order to find and assess the essential input parameter that affects the compressive strength of FA-GPC, a sensitivity comparison was carried out for the whole model [97]. The training dataset for the models was calculated by Solver in Excel. During the sensitivity analysis, several different training data sets were used. For each set, a single input variable was extracted at a time, and the effects of this variable were assessed by $R^2$, RMSE, MAE, OBJ, and SI, which is illustrated in Table 2. According to the obtained results, the curing temperature is the most

**Table 2. Sensitivity analysis using LRA, NLRA, and MLRA model.**

| | | LR Model | | | | | NLR Model | | | | | MLR Model | | | | |
|---|---|---|---|---|---|---|---|---|---|---|---|---|---|---|---|---|
| | | $R^2$ | RMSE | MAE | OBJ | SI | R2 | RMSE | MAE | OBJ | SI | R2 | RMSE | MAE | OBJ | SI |
| **Removed Parameter** | **None** | 0.84 | 4.65 | 3.76 | 3.09 | 0.15 | 0.86 | 4.19 | 3.35 | 2.71 | 0.14 | 0.79 | 5.09 | 3.95 | 3.40 | 0.17 |
| | *Si/Al* | 0.71 | 6.20 | 4.96 | 4.41 | 0.20 | 0.78 | 5.23 | 4.11 | 3.51 | 0.18 | 0.71 | 6.02 | 4.79 | 4.27 | 0.20 |
| | *l/b* | 0.73 | 6.01 | 4.75 | 4.21 | 0.20 | 0.86 | 4.23 | 3.33 | 3.22 | 0.18 | 0.79 | 5.10 | 3.95 | 3.41 | 0.17 |
| | *FA* (kg/m³) | 0.69 | 6.43 | 4.90 | 4.54 | 0.21 | 0.86 | 4.22 | 3.34 | 2.72 | 0.14 | 0.79 | 5.09 | 3.94 | 3.40 | 0.17 |
| | *F* (kg/m³) | 0.83 | 4.73 | 3.84 | 3.17 | 0.16 | 0.85 | 4.25 | 3.37 | 2.75 | 0.14 | 0.79 | 5.13 | 3.97 | 3.43 | 0.17 |
| | *C* (kg/m³) | 0.79 | 5.24 | 4.15 | 3.54 | 0.17 | 0.85 | 4.27 | 3.37 | 2.75 | 0.14 | 0.79 | 5.13 | 3.96 | 3.43 | 0.17 |
| | *SH* (kg/m³) | 0.79 | 5.27 | 4.21 | 3.58 | 0.17 | 0.85 | 4.24 | 3.37 | 2.75 | 0.14 | 0.79 | 5.11 | 3.97 | 3.42 | 0.17 |
| | *SS* (kg/m³) | 0.72 | 6.12 | 4.83 | 4.31 | 0.20 | 0.86 | 4.19 | 3.34 | 2.71 | 0.14 | 0.79 | 5.09 | 3.96 | 3.41 | 0.17 |
| | *SS/SH* | 0.84 | 4.65 | 3.78 | 3.10 | 0.15 | 0.86 | 4.20 | 3.36 | 2.72 | 0.14 | 0.79 | 5.12 | 3.98 | 3.43 | 0.17 |
| | *M* | 0.84 | 4.67 | 3.79 | 3.11 | 0.15 | 0.85 | 4.27 | 3.43 | 2.75 | 0.14 | 0.79 | 5.11 | 3.99 | 3.43 | 0.17 |
| | *T* (°C) | **0.40** | **8.90** | **7.00** | **7.67** | **0.29** | **0.43** | **8.43** | **6.71** | **7.11** | **0.28** | **0.36** | **8.88** | **6.98** | **7.84** | **0.29** |
| | *CD* (hr.) | 0.84 | 4.65 | 3.76 | 3.09 | 0.15 | 0.85 | 4.26 | 3.39 | 2.76 | 0.14 | 0.79 | 5.10 | 3.95 | 3.41 | 0.17 |
| | *A* (Day) | 0.75 | 5.70 | 4.47 | 3.92 | 0.19 | 0.76 | 5.40 | 4.23 | 3.65 | 0.18 | 0.71 | 6.01 | 4.67 | 4.21 | 0.20 |

significant variable for the prediction of the compressive strength of FA-GPC for the whole LR, NLR, and MLR models, and this is match with a variety of researches that have been performed in the literature [21,27,31,40,81,87,92]. In this study, the curing temperature for the obtained data was ranged from 23 to 120˚C, thus increasing the curing temperature considerably increased the compressive strength of FA-GPC. It is well documented in the literature that the compressive strength of FA-GPC is significantly affected by the curing temperature and duration. Longer curing time and curing at high temperature (50–100˚C) increases the compressive strength of FA-GPC, although the increase in strength may be insignificant for curing at more than 60˚C and for periods longer than 48 hrs. Therefore, for heat curing regimes, temperatures between 50–80˚C and curing time of 24 hr are widely accepted values used for a successful polymerization process. In addition, among the curing condition methods (oven, steam, and ambient), oven curing techniques have a better influence on the compressive strength of FA-GPC composites.

## 10. Conclusions

Predicting of compressive strength of FA-GPC by the reliable and accurate model can save time and cost. In this paper, linear regression (LR), nonlinear regression (NLR), and multi-logistic regression (MLR) were used to propose predictive models for the FBGC. Based on the 510 collected dataset from previous research works and the simulation of the compressive strength of the FA-GPC, the following conclusion can be drawn:

i. All the used models LR, NLR, and MLR could be successfully used to develop predictive models for the compressive strength of the FA-GPC. Overall, the NLR model has better performance than the other two models. The $R^2$ values for this model are 0.86, 0.75, and 0.79 for the training, testing, and validating datasets, respectively. In addition, other sensitivity indicators for the training dataset for the NLR model are 4.19 MPa, 3.35 MPa, 2.71, and 0.14 for the RMSE, MAE, OBJ, and SI, respectively.

ii. The $R^2$, RMSE, MAE, OBJ, and SI values were 0.84, 4.65MPa, 3.76MPa, 3.09, and 0.15, correspondingly, for the LR model for the training dataset. While these values are 0.79, 5.09 MPa, 3.95 MPa, 3.40, and 0.17, respectively, for the MLR model.

iii. The assessment and comparison of statistical parameters $R^2$, RMSE, MAE, OBJ, and SI for all the training, testing, and validating datasets validate the accuracy of the developed models properly.

iv. According to the sensitivity analysis approaches, the curing temperature, liquid to binder ratio, and sodium silicate content are the most effective independent variables for predicting the compressive strength of FA-GPC for all the models.

v. The eco-efficient fly ash-based geopolymer concrete studied here can participate in sustainable development because it is a cementless concrete and used industrial or agro by-product ashes as a binder material; these mixture properties lead to a reduction of the carbon dioxide percent in the air, energy consumption, as well as waste disposal and the cost of the construction.

## Author Contributions

**Conceptualization:** Ahmed Salih Mohammed, Azad A. Mohammed.

**Investigation:** Hemn Unis Ahmed.

**Validation:** Rabar H. Faraj.

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
