## [Decision Letter · Decision Letter 0]

17 Mar 2021

PONE-D-21-04168

Different mathematical approaches with SI and OBJ evaluations to predict the Compressive Strength of different mix proportions of Eco-efficient fly ash-based geopolymer concrete at various curing temperatures

PLOS ONE

Dear Dr. Mohammed,

Thank you for submitting your manuscript to PLOS ONE. After careful consideration, we feel that it has merit but does not fully meet PLOS ONE’s publication criteria as it currently stands. Therefore, we invite you to submit a revised version of the manuscript that addresses the points raised during the review process.

We look forward to receiving your revised manuscript.

Kind regards,

Tianyu Xie, Ph.D.

Academic Editor

PLOS ONE

Journal Requirements:

3. Thank you for stating the following financial disclosure: "NO"

4. Thank you for stating the following in your Competing Interests section:  "NO"

6. Please include your tables as part of your main manuscript and remove the individual files. Please note that supplementary tables (should remain/ be uploaded) as separate "supporting information" files

Reviewers' comments:

Reviewer's Responses to Questions

**Comments to the Author**

1. Is the manuscript technically sound, and do the data support the conclusions?

Reviewer #1: Yes

Reviewer #2: Partly

Reviewer #3: Yes

2. Has the statistical analysis been performed appropriately and rigorously? 

Reviewer #1: Yes

Reviewer #2: Yes

Reviewer #3: Yes

3. Have the authors made all data underlying the findings in their manuscript fully available?

Reviewer #1: Yes

Reviewer #2: Yes

Reviewer #3: Yes

4. Is the manuscript presented in an intelligible fashion and written in standard English?

Reviewer #1: Yes

Reviewer #2: No

Reviewer #3: Yes

5. Review Comments to the Author

Reviewer #1: This paper presents linear, non-linear, and multi-logistic regression models to predict the compressive strength of fly ash-based geopolymer concrete (FA-GP concrete) using 510 mixes collected from the literature. The content is interesting but this reviewer believes there are several critical concerns need to be carried out. Therefore, this reviewer recommends publication of this paper provided that the following major revisions are successfully carried out.

1. The title of this paper does not fit with its content. The title is lengthy and it should be shortened. Please also do not use abbreviations in the title of paper.

2. Author should briefly discuss their innovation in Abstract

3. The authors used the linear, non-linear, and multi-logistic regression models to predict the compressive strength of fly ash-based geopolymer concrete. The authors should clearly explain why they chose these three prediction models in details. Please also explain the advantages and disadvantages of these three prediction models over other prediction models.

4. As authors say: “In this regard, a comprehensive dataset consists of 510 samples were collected in several academic research studies and analyzed to develop the models.”

Authors should clearly explain why these 510 mixes are comprehensive enough for their prediction models. Please also explain what criteria have been considered for collecting these 510 mixes from previous studies.

5. Authors should clearly explain on what basis they chose these twelve input variables for the prediction of compressive strength of FA-GP concrete as there are similarities between some input variables

6. Please choose a better abbreviation for fly ash-based geopolymer concrete (e.g. FA-GPC)

7. Introduction is lengthy with unnecessary explanations on geopolymer concrete behavior. Please shorten Introduction and rewrite it with a more focus on literature review on the use of different machine learning methods for prediction of mechanical properties of concrete.

8. In Introduction, please explain the abbreviation for GGBFS.

9. There should be more discussion on results not just providing the results. Please compare the results obtained by the proposed regression models with other machine learning methods such as artificial neural networks, genetic programming or other regressions models and clearly discuss the results.

10. On what basis, 66%, 17% and 17% of datasets were chosen respectively for training, testing and validating? There should be a sensitivity analysis for the discretization of dataset for training, testing and validating. If there is, please clearly discuss the results.

11. As authors say: “It can be noticed from both figures that the predicted and measured values of compression strength are closer for the NLR mode, which indicates the superior performance of the NLR model compared to other models.”

Please clearly explain why NRL has a better performance than the two other models.

12. As authors say: “According to the obtained results, the curing temperature is the most significant variable for the prediction of the compression strength of FBGC for the whole LR, NLR, and MLR models and this is match with a variety of researches that have been performed in the literature”

Please clearly explain why this happens?

To discuss the importance of curing temperature for compressive strength prediction, please discretize your dataset based on the curing method (ambient and oven curing) and discuss the results for (i) ambient curing only; (ii) oven curing only and (iii) ambient and oven curing as you have already done.

13. There are many grammatical errors that need to be corrected

Reviewer #2: This paper presented a regression model for predicting the compressive strength of geopolymer concrete. A comprehensive database including 510 samples was established and used for developing/validating the proposed model. Although this research is very straightforward, it contributes to guiding the mix formulation design for geopolymer concrete. The following comments have to be addressed before it can be considered for publication.

(1) The title should be rephrased/simplified to highlight the key research topic in this paper.

(2) General: The development and validation of the regression model are very straightforward. The novelty of this research should be highlighted somewhere in the paper.

(3) General: The industry or agro by-product ashes were usually used as the precursor in geopolymer instead of binder. It is suggested that the authors check throughout the paper.

(4) Abstract: It stated that compressive strength of concrete is the most important mechanical property of concrete. The authors may justify why the compressive strength is the most important property or rephrase this statement.

(5) Abstract: “in the construction field at least 28 days is required until the compressive strength results are available.” This statement is not clear as well. In the construction site, it is not necessary to wait for 28 days for concrete hardening before starting other construction activities. 28-day compressive strength is adopted as the representative strength of concrete. It may not be able to alter the construction process.

(6) Abstract: what do the authors mean by “elimination of framework elements”?

(7) Abstract: the authors claimed that the developed model can be used to guide the construction process. However, the model included the curing duration inside ovens as one of the parameters, which seems to be impractical to provide oven curing during the construction process. This has to be justified in the development of the prediction model.

(8) The abstract needs to be improved through highlighting the key findings of this paper.

(9) Methodology: The datasets were divided into three groups for different purposes. How did the authors categorize the collected data? How did the authors decide the size of each group?

(10) Methodology: the information about the ranges of each parameter in Research Significance and Methodology is duplicated. You may remove this information in the Research significance section.

(11) Statistical assessment: A statistical analysis was conducted to check the relationship between the parameters and compressive strength. As the compressive strength of geopolymer concrete is affected by more than one parameter, these relationships can tell limited useful information in the development of prediction model. The authors may clearly justify the reason to include this section.

(12) The use of English should be checked throughout the manuscript as there are some grammatical errors or typos.

Reviewer #3: abstract must include numerical findings of the study

"compression strength" should be rerplaced with compreesive strength, please check all manuscript

standard deviation and variance of the test data selected from those 500 dataset must be calculated and provided in revised version

6. PLOS authors have the option to publish the peer review history of their article (what does this mean?). If published, this will include your full peer review and any attached files.

Reviewer #1: No

Reviewer #2: No

Reviewer #3: No

---

## [Author Response · Author response to Decision Letter 0]

4 May 2021

Response Letter

Title: Different mathematical approaches with SI and OBJ evaluations to predict the Compressive Strength of different mix proportions of Eco-efficient fly ash-based geopolymer concrete at various curing temperatures 

Dear Editor: 

Dear Managing Editor and co-Editors in Chief,

We appreciate the time and contributions of the editor and referees in reviewing this manuscript. We revised the whole manuscript based on your comments. We have addressed all issues indicated in the review report and believed that the revised version could meet the journal publication requirements. In the revised manuscript, the changes are highlighted below. 

The responses to the comments of reviewers can be seen in the following.

Answer to Reviewer #1:

Many thanks for your valuable comments. Regarding your precious technical points, the manuscript revised parts have been highlighted in yellow color. The authors agree with your essential comments, and the manuscript was significantly improved according to your comments. 

C-1: The title of this paper does not fit with its content. The title is lengthy and it should be shortened. Please also do not use abbreviations in the title of paper.

Answer: 

Thanks for this valuable comment; based on this comment and another comment from reviewer No.2, the ttile of the paper has been modified to Different mathematical approaches with SI and OBJ evaluations to predict the Compressive Strength of different mix proportions of Eco-efficient fly ash-based geopolymer concrete at various curing temperatures to (Systematic multiscale models to predict the compressive strength of fly ash-based geopolymer concrete at various mixture proportions and curing regimes. 

C-2: Author should briefly discuss their innovation in Abstract

Answer: 

The study's novelty was shown further, actually according to the extensive review made by the authors of this study on a related matter, despite the wide application of geopolymer concrete, a reliable model to the use of geopolymer concrete to be used by the construction industry is very scarce. Most of the attempts have been related to a single scale model without covering comprehensive experimental data or modeling. Thus, the effect of several parameters such as effective parameters on the compressive strength of the fly ash-based geopolymer concrete, including SiO2/Al2O3 (Si/Al) of fly ash binder, alkaline liquid to binder ratio (l/b), fly ash (FA) content, fine aggregate (F) content, coarse aggregate (C) content, sodium hydroxide (SH)content, sodium silicate (SS) content, (SS/SH), molarity (M), curing temperature (T), curing duration inside ovens (CD) and specimen ages (A) were quantified. We address this issue in the manuscript as highlighted with the Green colour in the revised manuscript.

C-3: The authors used the linear, non-linear, and multi-logistic regression models to predict the compressive strength of fly ash-based geopolymer concrete. The authors should clearly explain why they chose these three prediction models in details. Please also explain the advantages and disadvantages of these three prediction models over other prediction models.

Answer: 

Advantages of Linear Regression

Linear Regression is a straightforward algorithm that can be implemented very quickly to give satisfactory results. Furthermore, these models can be trained rapidly and efficiently even on systems with relatively low computational power when compared to other complex algorithms. Linear regression has a considerably lower time complexity when compared to some of the different machine learning algorithms. The mathematical equations of Linear regression are also reasonably easy to understand and interpret. Hence Linear regression is straightforward to master.

Linear regression fits linearly separable datasets almost perfectly and is often used to find the nature of the relationship between variables.

Overfitting can be reduced by regularization

Overfitting is a situation that arises when a machine learning model fits a dataset very closely and hence captures the noisy data as well. This negatively impacts the performance of the model and reduces its accuracy on the test set.

Regularization is a technique that can be easily implemented and can effectively reduce the complexity of a function to reduce the risk of overfitting.

Disadvantages of Linear Regression

Underfitting: A situation that arises when a machine learning model fails to capture the data adequately.This typically occurs when the hypothesis function cannot fit the information well. Since linear regression assumes a linear relationship between the input and output variables, it fails to fit complex datasets properly. In most real-life scenarios, the relationship between the dataset variables isn't linear, and hence a straight line doesn't fit the data correctly. In such situations, a more complex function can capture the data more effectively. Because of this, most linear regression models have low accuracy.

Advantage of NLR

This method offers an entirely different approach for dealing with the non-linear model and the slowly time-varying or uncertain parameters of the system.

Disadvantage of NLR

(1) Difficulty in finding the Lyapunov functions. 

(2) Complexity in the integration of non-linear observer with HVAC.

 (3) Sensitivity to parameter variation.

 (4) Limited operating range in state feedback.

 (5) Proof of stability is challenging. 

(6) Need for measuring all state variables or additional measurement.

 (7) Possibility only on stable processes.

 (8) The non-linear observer has required if the all-state variables were not measurable.

Advantage and Disadvantage of MLR

Advantages Disadvantages

Logistic regression is easier to implement, interpret, and very efficient to train. If the number of observations is lesser than the number of features, Logistic Regression should not be used. Otherwise, it may lead to overfitting.

It makes no assumptions about distributions of classes in feature space. It constructs linear boundaries.

It can easily extend to multiple classes(multinomial regression) and a natural probabilistic view of class predictions. The major limitation of Logistic Regression is linearity between the dependent variable and the independent variables.

It provides a measure of how appropriate a predictor(coefficient size)is and its direction of association (positive or negative). It can only be used to predict discrete functions. Hence, the dependent variable of Logistic Regression is bound to the discrete number set.

It is very fast at classifying unknown records. Non-linear problems can’t be solved with logistic regression because it has a linear decision surface. Linearly separable data is rarely found in real-world scenarios.

C-4: As authors say: “In this regard, a comprehensive dataset consists of 510 samples were collected in several academic research studies and analyzed to develop the models.” Authors should clearly explain why these 510 mixes are comprehensive enough for their prediction models. Please also explain what criteria have been considered for collecting these 510 mixes from previous studies.

Answer: 

As you know that there is a wide range of data regarding geopolymer concrete with different base source materials, including silica fume (SF), ground granulated blast furnace slag (GGBFS), fly ash (FA), rice husk ash (RHA), Metakaolin (MK), red mud (RM) and so on. But in this study only the studies which were used fly ash as a based geopolymer have been taken into account for the analysis. Therefore, the authors tried to find out a wide range of papers that includes the required parameters that we have in the input parameters for developing the models. For instance, if there is research in the literature that focuses on fly ash-based geopolymer concrete, but they don’t provide their mix proportions or any other input model parameters, the paper was taken out from the database. Finally, based on this accurate comment, the authors decided to clarify this issue in the revised manuscript. 

C-5: Authors should clearly explain on what basis they chose these twelve input variables for the prediction of compressive strength of FA-GP concrete as there are similarities between some input variables

Answer:

Thanks for this valuable comment. The authors have this clarification regarding this point, and we hope it satisfies the reviewer expectations: The authors agree with this helpful comment, and actually, there is no direct relationship between the individual parameters with the compressive strength of fly ash-based geopolymer concrete, as shown from figure 2 to figure 13. This is because that the compressive strength is affected by more than one parameter. Therefore, the authors decided to show the effect of a wide range of mixed proportion variables and different curing regimes on the compressive strength of fly ash-based geopolymer concrete. However, based on this valuable comment, we addressed this issue in the revised manuscript. 

From the developed models in this paper, the researchers will be able to predict the compressive strength of the geopolymer concrete (Y) by using the including SiO2/Al2O3 (Si/Al) of fly ash binder, alkaline liquid to binder ratio (l/b), fly ash (FA) content, fine aggregate (F) content, coarse aggregate (C) content, sodium hydroxide (SH) content, sodium silicate (SS) content, (SS/SH), molarity (M), curing temperature (T), curing duration as an (X) values with a high degree of accuracy. 

C-6: Please choose a better abbreviation for fly ash-based geopolymer concrete (e.g. FA-GPC).

Answer:

Based on this comment, the FBGC to FA-GPC in the whole revised manuscript has been changed.

C-7: Introduction is lengthy with unnecessary explanations on geopolymer concrete behavior. Please shorten Introduction and re-write it with a more focus on literature review on the use of different machine learning methods for prediction of mechanical properties of concrete.

Answer: 

The introduction section is updated and shortened in the revised version.

C-8: In Introduction, please explain the abbreviation for GGBFS.

Answer: 

This abbreviation is used for Ground Granulated Blast Furnace Slag (GGBFS). However, sometimes GGBS is used instead of GGBFS. Based on this comment, we add the full name of this abbreviation in the introduction part. 

C-9: There should be more discussion on results not just providing the results. Please compare the results obtained by the proposed regression models with other machine learning methods such as artificial neural networks, genetic programming or other regressions models and clearly discuss the results.

Answer:

The authors tried to re-write the results with an in-depth discussion in the revised manuscript according to this valuable comment. Agree with comment reading to use and compare the results with machine learning methods such as artificial neural networks, genetic programming, M5Ptree …etc. Which are ongoing Ph.D. study and only these three models were considered in this study. 

C-10: On what basis, 66%, 17% and 17% of datasets were chosen respectively for training, testing and validating? There should be a sensitivity analysis for the discretization of dataset for training, testing and validating. If there is, please clearly discuss the results.

Answer: 

As it is very well known that we take 510 datasets from the literature, and also based on the information in the litersture [Salih, A., Rafiq, S., Mahmood, W., Ghafor, K., & Sarwar, W. (2021). Various simulation techniques to predict the compressive strength of cement-based mortar modified with micro-sand at different water-to-cement ratios and curing ages. Arabian Journal of Geosciences, 14(5), 1-14.] the majority (2/3) of these data to create the models, and we described it as the training data, and the remained data (1/3) were used to test and validate the developed modes [43]. 

C-11: As authors say: “It can be noticed from both figures that the predicted and measured values of compression strength are closer for the NLR mode, which indicates the superior performance of the NLR model compared to other models.” Please clearly explain why NRL has a better performance than the two other models.

Answer:

In the mathematic increasing, the model variable the accuracy of the equation will be increased. Since the NLR model has a larger equation constant than the LR and MLR model, so the RMSE and MAE of NLR are lower than the other two models. 

C-12: As authors say: “According to the obtained results, the curing temperature is the most significant variable for the prediction of the compression strength of FBGC for the whole LR, NLR, and MLR models and this is match with a variety of researches that have been performed in the literature” Please clearly explain why this happens? To discuss the importance of curing temperature for compressive strength prediction, please discretize your dataset based on the curing method (ambient and oven curing) and discuss the results for (i) ambient curing only; (ii) oven curing only and (iii) ambient and oven curing as you have already done.

Answer:

The authors agree with this valuable comment. Therefore, the authors provide more discussion on the effect of curing regimes on the compressive strength of fly ash-based geopolymer concrete in the revised manuscript version. In addition, the authors tried to do this study for the researches that use only the same curing conditions. During the sensitivity analysis, several different training data sets were used, and for each group, a single input variable was extracted at a time, and the effects of this variable were assessed by R2, RMSE, MAE, OBJ, and SI, which is illustrated in Table 2. In the study, reliable multivariable models with single curing regimes using a systematic multiscale and other machine learning techniques such as artificial neural network models are ongoing study to propose efficient models to predict the mechanical properties of fly ash-based geopolymer concrete at a single curing condition.

C-13: There are many grammatical errors that need to be corrected

Answer:

The whole paper, including text, tables, and graphs, has been modified in terms of grammar errors. Thanks. 

Answer to Reviewer #2:

C-1: The title should be re-phrased/simplified to highlight the key research topic in this paper.

Answer: 

The title has been modified. 

C-2: General: The development and validation of the regression model are very straightforward. The novelty of this research should be highlighted somewhere in the paper.

Answer: 

According to this comment and another comment from reviewer No.1 which also related to this comment, the novelty of the study was shown further, actually according to the extensive review made by the authors of this study on a related matter, despite the wide application of geopolymer concrete, a reliable model to the use of geopolymer concrete to be used by the construction industry is very scarce. Most of the attempts have been related to a single scale model without covering a wide experimental data or multiple parameters. Thus, the effect of several parameters such as effective parameters on the compressive strength of the fly ash-based geopolymer concrete, including SiO2/Al2O3 (Si/Al) of fly ash binder, alkaline liquid to binder ratio (l/b), fly ash (FA) content, fine aggregate (F) content, coarse aggregate (C) content, sodium hydroxide (SH)content, sodium silicate (SS) content, (SS/SH), molarity (M), curing temperature (T), curing duration inside ovens (CD) and specimen ages (A) were quantified. We address this issue in the manuscript as highlighted with the Green colour in the revised manuscript.

C-3: General: The industry or agro by-product ashes were usually used as the precursor in geopolymer instead of the binder. It is suggested that the authors check throughout the paper.

Answer: 

Thanks for this valuable and essential comment, we consider this an accurate analysis, and we check it throughout the revised manuscript.

C-4: Abstract: It stated that compressive strength of concrete is the most important mechanical property of concrete. The authors may justify why the compressive strength is the most important property or rephrase this statement.

Answer:

Based on reference No. 18 [Neville, A. M., & Brooks, J. J. (2010). Concrete technology], and Mohammed, A., Burhan, L., Ghafor, K., Sarwar, W., & Mahmood, W. (2020). Artificial neural network (ANN), M5P-tree, and regression analyses to predict the early age compression strength of concrete modified with DBC-21 and VK-98 polymers. Neural Computing and Applications, 1-23.] compressive strength is the most important mechanical property of all concrete composites, including fly ash-based geopolymer concrete. Compressive strength gives a general performance about the quality of the concrete composites. Also, with increasing the compressive strength of the concrete the rest of the mechanical properties such as tensile, flexural, and bonding strengths will increase. 

C-5: Abstract: “in the construction field at least 28 days is required until the compressive strength results are available.” This statement is not clear as well. In the construction site, it is not necessary to wait for 28 days for concrete hardening before starting other construction activities. 28-day compressive strength is adopted as the representative strength of concrete. It may not be able to alter the construction process.

Answer: 

The authors agree with this accurate comment, so this point was modified as highlighted in the revised manuscript. Actually, we want to say that, by using the developed models, you can predict or estimate the compressive strength of fly ash-based geopolymer concrete. It is essential for the construction process and the structural design of the concrete composites.

C-6: Abstract: what do the authors mean by “elimination of framework elements”?

Answer: 

We use this statement “elimination of framework elements” as synonyms of “removal of formworks.” So, we change the information as highlighted in the revised manuscript. In the field, removal of formworks is based on the strength gain of the concrete and some other parameters, so if we know the mixture ingredients with the curing regime, we can predict the strength gain with time by applying these developed models. As a consequence, this developed model is beneficial for this issue. 

C-7: Abstract: the authors claimed that the developed model can be used to guide the construction process. However, the model included the curing duration inside ovens as one of the parameters, which seems to be impractical to provide oven curing during the construction process. This has to be justified in the development of the prediction model.

Answer: 

 In the construction process, the use of fly ash-based geopolymer concrete is limited to the precast elements and some field constructions in the ambient curing conditions. So, it is easy to predict the fly ash-based geopolymer concrete precast elements' compressive strength in the controlled curing conditions. However, from the construction point of view for the field construction, it may be difficult to control the ambient curing conditions; so, it is suggested to take the average temperature degree during the 24 hrs to be used inside the developed models to predict the compressive strength of the fly ash-based geopolymer concrete. Finally, we consider this valuable comment in the revised manuscript.

C-8: The abstract needs to be improved through highlighting the key findings of this paper.

Answer: 

The abstract section was modified, and we consider this critical comment in the revised manuscript version as highlighted in the paper. 

C-9: Methodology: The datasets were divided into three groups for different purposes. How did the authors categorize the collected data? How did the authors decide the size of each group?

Answer: 

The authors have this clarification regarding this point. We hope it satisfies the reviewer's expectations: the 510 datasets from the literature were used in this study. We decide to use the majority (2/3) of these data to create the models, and we described it as the training data, and the remained data (1/3) were used to test and validate the developed modes. Actually we follow the reference No. 43 (Golafshani, E. M., Behnood, A., & Arashpour, M. (2020). Predicting the compressive strength of normal and High-Performance Concretes using ANN and ANFIS hybridized with Grey Wolf Optimizer. Construction and Building Materials, 232, 117266.) Who they divided their datasets into three groups, training, testing, and validating as we did it. 

C-10: Methodology: the information about the ranges of each parameter in Research Significance and Methodology is duplicated. You may remove this information in the Research significance section.

Answer: 

The authors agree with the reviewer comment, so we decided to delete the information about the range of each parameter in the Research Significant part in the revised manuscript. 

C-11: Statistical assessment: A statistical analysis was conducted to check the relationship between the parameters and compressive strength. As the compressive strength of geopolymer concrete is affected by more than one parameter, these relationships can tell limited useful information in the development of prediction model. The authors may clearly justify the reason to include this section.

Answer: 

The authors agree with this valuable comment. No direct relationship between the individual parameters with the compressive strength of fly ash-based geopolymer concrete was observed, as shown from Fig. 2 and Fig. 13. This is because that the compressive strength is affected by more than one parameter (as you mention). Also, this issue is one reason that made us decide to do this study to show the problem discussed above and input multi-parameters that influence the compressive strength of fly ash-based geopolymer concrete. The authors mention this issue in the Modeling part as highlighted in the revised manuscript.

C-12: The use of English should be checked throughout the manuscript as there are some grammatical errors or typos.

Answer: 

The whole paper, including text, tables, and graphs, has been modified in terms of grammar errors. Thanks. 

Answer to Reviewer #3:

C-1: Abstract must include numerical findings of the study.

Answer: 

The authors agree with this valuable comment; the abstract section was updated to satisfy this critical point mentioned by the reviewer, as highlighted in the revised manuscript 

C-2: "compression strength" should be replaced with compressive strength, please check all the manuscripts.

Answer: 

Based on this comment, the Compression strength to compressive strength in the whole revised manuscript

C-3: Standard deviation and variance of the test data selected from that 500 dataset must be calculated and provided in revised version.

Answer: 

The standard deviation and variance are provided for the whole datasets in the revised manuscript, and it can be seen from figure 3 to figure 13 in the revised manuscript.

Thank you for your time and kind consideration.

Best Regards,

Corresponding Author

---

## [Decision Letter · Decision Letter 1]

27 May 2021

Systematic multiscale models to predict the compressive strength of fly ash-based geopolymer concrete at various mixture proportions and curing regimes

PONE-D-21-04168R1

Dear Dr. Mohammed,

We’re pleased to inform you that your manuscript has been judged scientifically suitable for publication and will be formally accepted for publication once it meets all outstanding technical requirements.

Kind regards,

Tianyu Xie, Ph.D.

Academic Editor

PLOS ONE

Additional Editor Comments (optional):

Reviewers' comments:

Reviewer's Responses to Questions

**Comments to the Author**

1. If the authors have adequately addressed your comments raised in a previous round of review and you feel that this manuscript is now acceptable for publication, you may indicate that here to bypass the “Comments to the Author” section, enter your conflict of interest statement in the “Confidential to Editor” section, and submit your "Accept" recommendation.

Reviewer #1: All comments have been addressed

Reviewer #3: All comments have been addressed

2. Is the manuscript technically sound, and do the data support the conclusions?

Reviewer #1: Yes

Reviewer #3: Yes

3. Has the statistical analysis been performed appropriately and rigorously? 

Reviewer #1: Yes

Reviewer #3: Yes

4. Have the authors made all data underlying the findings in their manuscript fully available?

Reviewer #1: Yes

Reviewer #3: Yes

5. Is the manuscript presented in an intelligible fashion and written in standard English?

Reviewer #1: Yes

Reviewer #3: Yes

6. Review Comments to the Author

Reviewer #1: (No Response)

Reviewer #3: In my opinion the authors are addressed all my previous comments in their revised manuscrpt. The paper is now appropriate for publication

7. PLOS authors have the option to publish the peer review history of their article (what does this mean?). If published, this will include your full peer review and any attached files.

Reviewer #1: No

Reviewer #3: **Yes: **Ertug Aydin

---

## [Editor Report · Acceptance letter]

1 Jun 2021

PONE-D-21-04168R1 

Systematic multiscale models to predict the compressive strength of fly ash-based geopolymer concrete at various mixture proportions and curing regimes 

Dear Dr. Mohammed:

I'm pleased to inform you that your manuscript has been deemed suitable for publication in PLOS ONE. Congratulations! Your manuscript is now with our production department. 

Kind regards, 

on behalf of

Dr. Tianyu Xie 

Academic Editor

PLOS ONE